# WEAK-TO-STRONG GRAPHRAG: ALIGNING WEAK RETRIEVERS TO LARGE LANGUAGE MODELS FOR GRAPH-BASED RETRIEVAL AUGMENTED GENERATION

## ABSTRACT

Graph-based retrieval-augmented generation (RAG) enables large language models (LLMs) to ground responses with structured external knowledge from up-to-date knowledge graphs (KGs) and reduce hallucinations. However, LLMs often rely on a *weak retriever* in graph-based RAG: I) Due to the lack of ground truth, the retriever is often trained on *weak supervision*, which often introduces *spurious signals* to the LLMs. II) Due to the abstraction of graph data, the retrieved knowledge is often presented in *unorganized* forms. To mitigate the issue, we present `Refined graph-based RAG (ReG)`, which refines the weak supervision for graph-based RAG. Specifically, `ReG` incorporates LLM feedback to get rid of spurious signals and improve the quality of the supervision. Meanwhile, `ReG` also uses a structure-aware reorganization module to refactor the retrieval results into logically coherent evidence chains. Experiments on prominent benchmarks demonstrate that `ReG` significantly and consistently brings improvements across different LLM backbones by up to 10%. The improved supervision quality enables `ReG` to match the state-of-the-art performance with 5% training data and to transfer to out-of-distribution KGs. Notably, when adopted to reasoning-based LLMs, `ReG` reduces the reasoning token cost by up to 30% and improves the performance by up to 4%.

## 1 INTRODUCTION

Large language models (LLMs) have achieved remarkable success and continue to advance (Bang et al., 2023; Raiaan et al., 2024; Yang et al., 2024). In particular, long-context LLMs (Liu et al., 2025; Huang et al., 2023; Zhao et al., 2023) extend context windows to millions of tokens (Team et al., 2024), and large reasoning models (LRMs) like OpenAI o3 (OpenAI, 2025) and DeepSeek R1 (Guo et al., 2025a) further push the frontier in extremely complicated tasks. However, even frontier LLMs fall short in processing extremely lengthy or noisy input contexts: long-context LLMs often fail to effectively extract the useful information from the long context (Hsieh et al.; Liu et al., 2024c), while LRMs tend to spend excessive reasoning costs on irrelevant or redundant contents (Chen et al., 2024b; Sui et al., 2025; Feng et al., 2025). Therefore, it is essential to retrieve the desired information for LLMs, especially when handling complicated multi-hop reasoning tasks (Han et al., 2024).

Retrieval-augmented generation (RAG) has emerged as a powerful paradigm for enhancing LLMs (Borgeaud et al., 2022; Gao et al., 2023; Chen et al., 2024a; Fan et al., 2024). By retrieving query-relevant information from external knowledge sources, RAG reduces the hallucinations of LLMs (Ji et al., 2023; Huang et al., 2025; Liu et al., 2024b), factuality (Dhingra et al., 2022; Kasai et al., 2023), and specificity of LLM responses (Li et al., 2023a). Recently, graph-based RAG extended text-based RAG to further enhance retrieval effectiveness in capturing the interconnections between facts. By retrieving over graph-structured textual and document-level knowledge from a knowledge graph (KG), graph-based RAG provides more structure-aware, interpretable, and compositional information to text-based retrieval, particularly effective for handling complex multi-hop reasoning tasks (Chein & Mugnier, 2008; Robinson et al., 2015; Peng et al., 2024; Han et al., 2024).

Despite the promise, graph-based RAG relies on *weak retrievers*, *misaligned* with LLMs: **(I) weak supervision**: Different from text-based RAG, there does not exist a general-purpose structural information retriever (Luo et al., 2025b), and the ground truth of graph-based RAG is usually lacking.

Figure 1: Overall framework of ReG. Given a KG $\mathcal{G}$ with a query-answer pair $(q, a)$, ReG first constructs a candidate path pool $\mathcal{P}$ to cover diverse candidate reasoning paths. Then, ReG uses LLM to select high-quality $\widehat{\mathcal{G}}^+$ for training retrievers. During inference, retrieved items are reorganized into logic-consistent chains to better align with the needs for LLM reasoning.

Hence, graph-based retriever often requires training on specific datasets with *heuristic-based weak supervision signals* (Zhang et al., 2022; Luo et al., 2023). However, the weak supervision signals can often miss key supporting evidence or include spurious connections unrelated to the reasoning logic. Especially when the query requires multi-hop reasoning over KGs, missing key intermediate steps in the supervision signals will severely limit the performance of the retriever. **(II) misorganized representation**: The retrieved graph information can be represented in a variety of forms (Mavromatis & Karypis, 2024; Luo et al., 2023; Li et al., 2024a), and orders. Nevertheless, LLMs are generically sensitive to the ordering of context (Chen et al., 2024c; Guo et al., 2025c). The misorganized representation further adds to the complexity of the graph-based RAG and raises the question:

*How can one align the weak retrievers to LLMs in graph-based RAG?*

To mitigate the issue, we present Refined graph-based RAG (ReG), which incorporates the rich knowledge of LLMs to refine and align the weak supervision in graph-based RAG. Essentially, we show that graph-based RAG can be considered as a black-box combinatorial search over the KG $\mathcal{G}$: given a query $q$, the goal is to identify a minimal sufficient subgraph $\widehat{\mathcal{G}}^* \subseteq \mathcal{G}$ for an LLM to answer $q$ correctly. Here, the LLM serves as a black-box evaluator that assesses the utility of retrieved subgraphs. Using the formulation, we show that resolving the original black-box optimization problem is computationally intractable and thus not feasible under realistic LLM usage budgets. Therefore, ReG incorporates the LLMs in a simple yet effective way: using LLMs to select better reasoning chains among the candidate chains extracted from KG. The improved supervision signal improves the identification of the optimal subgraph in a cost-efficient manner. To align the retrieval results to LLMs, ReG reorganizes the retrieved contents into logic-preserving chains, which is simple but significantly mitigates distraction and inefficiency during LLM reasoning.

Extensive experiments demonstrate ReG achieves state-of-the-art results on prominent multi-hop knowledge graph question answering (KGQA) benchmarks. Notably, it yields retrievers with stronger zero-shot generalizability to out-of-distribution (OOD) KGs, mitigating the weakness of lacking foundation models in graph-based RAG. The improved supervision enables ReG to match the state-of-the-art performance with 5% training data. ReG also brings measurable gains with reduced token costs up to 30% when paired with the frontier reasoning LLMs. These results highlight the effectiveness, data-efficiency, and versatility of ReG as a generally applicable framework that enhances reasoning quality even under the most capable frontier LLMs.

## 2 PRELIMINARIES AND RELATED WORKS

**Preliminaries.** In this work, we focus on **knowledge graph question answering** (KGQA), a central task in graph-based RAG: Given a natural language query $q$ and a KG $\mathcal{G}$, the LLM answers the query by reasoning over the retrieved subgraph $\widehat{\mathcal{G}} \subseteq \mathcal{G}$, which ideally captures all critical supporting evidence while minimizing distracting noise. A **knowledge graph** is a structured representation of

factual knowledge, typically formulated as a collection of triples $\mathcal{G} = \{(h, r, t) \mid h, t \in \mathcal{E}, r \in \mathcal{R}\}$, where $\mathcal{E}$ and $\mathcal{R}$ denote the sets of entities and relations, respectively. Each triple $\tau = (h, r, t)$ represents a directed relation $r$ from head entity $h$ to tail entity $t$. Given the query $q$, we denote by $\mathcal{E}_q \subseteq \mathcal{E}$ the set of **query entities** extracted from $q$, and by $\mathcal{A}_q \subseteq \mathcal{E}$ the corresponding set of **answer entities**. A **reasoning path** is defined as an ordered sequence of connected triples $P = (\tau_1, \dots, \tau_k)$, where each $\tau_i = (h_i, r_i, t_i) \in \mathcal{G}$ satisfies $t_i = h_{i+1}$ for all $1 \leq i < k$.

Due to the space limits, we discuss the most relevant work below and leave a more detailed discussion of related work in Appendix C.

**Graph-based Retrieval.** Due to the lack of an oracle in graph-based RAG, it often relies on heuristic priors or weak supervisions to guide the retrieval, which can be broadly categorized as: *Training-free* methods typically adopt graph-based heuristics (Gutiérrez et al., 2024; 2025) or LLM-guided stepwise exploration over the graph (Gu et al., 2022; Sun et al., 2023; Xiong et al., 2024). However, graph algorithms often underperform as they struggle to effectively leverage semantic and structural information (Luo et al., 2025b;a). *Training-based* methods typically train a parametric retriever to recognize critical substructure using weak supervisions such as query-answer shortest paths (Zhang et al., 2022; Luo et al., 2023), which may omit critical intermediate evidence and introduce spurious connections (Li et al., 2024a). Differently, we aim to bridge the gap by integrating the rich knowledge of LLMs to improve the quality of weak supervision for retriever training.

**RAG with LLM Feedback.** Recent efforts in text-based RAG have explored using LLM feedback to directly optimize the retriever capability. Shi et al. (2023) aligns retriever output distributions with the perplexity-reducing behavior of LMs. Li et al. (2024b) iteratively tunes the LLM generator and retriever by aligning data preferences between both modules. Han et al. (2025) trains an intermediary module by collecting preference data from both the LLM and the retriever to enhance RAG performance. While effective for document-level retrieval, in graph-based RAG, the combinatorial explosion of subgraph candidates and their intricate dependencies pose unique challenges.

# 3 DRAWBACKS OF WEAK RETRIEVERS IN GRAPH-BASED RAG

## 3.1 GRAPH-BASED RAG AS BLACK-BOX COMBINATORIAL OPTIMIZATION

The graph-based RAG can be considered as a black-box combinatorial optimization problem. Given the query $q$ for a KG $\mathcal{G}$ with $N$ triples, let $s(\cdot, q) : \mathcal{G} \mapsto [0, 1]$ be an unknown scoring function assigning relevance scores to items, where $s(\tau, q) > 0$ indicates $\tau$ is relevant to $q$. The retriever aims to find the underlying *unknown oracle set* $\widehat{\mathcal{G}}^* \subseteq \mathcal{G}$, defined as $\widehat{\mathcal{G}}^* := \{\tau \in \mathcal{G} \mid s(\tau, q) > 0\}$, and assumed to be sparse, *i.e.*, $|\widehat{\mathcal{G}}^*| \sim \mathcal{O}(1) \ll |\mathcal{G}|$, which enables LLMs to answer the question correctly. Hence, LLMs can be considered as the black-box evaluator of the retrieved KG subgraph $\widehat{\mathcal{G}}$.

**Definition 3.1 (Black-box LLM evaluator)** *A black-box reward function $r(\cdot, q) : 2^{\mathcal{G}} \mapsto \mathbb{R}$ evaluates how well a subset $\widehat{\mathcal{G}} \subseteq \mathcal{G}$ aligns with the oracle set $\widehat{\mathcal{G}}^*$ by the following properties:*

*i) Aggregation. The reward aggregates relevance scores $s(\tau, q)$ for $\tau \in \widehat{\mathcal{G}} \cap \widehat{\mathcal{G}}^*$ and penalties $\delta(\tau, q) \geq 0$ for $\tau \in \widehat{\mathcal{G}} \backslash \widehat{\mathcal{G}}^*$. Formally, $r(\widehat{\mathcal{G}}, q) := f\left(\{s(\tau, q) \mid t \in (\widehat{\mathcal{G}} \cap \widehat{\mathcal{G}}^*)\}, \{\delta(\tau, q) \mid \tau \in (\widehat{\mathcal{G}} \backslash \widehat{\mathcal{G}}^*)\}\right)$, where $\delta$ quantifies the LLM evaluator's robustness to noise: $\delta(\tau, q) = 0$ means full robustness, while $\delta(\tau, q) > 0$ imposes a penalty on $r(\widehat{\mathcal{G}}, q)$.*

*ii) Exactness. For any $\widehat{\mathcal{G}} \subset \mathcal{G}$, the reward reaches its maximum if and only if $\widehat{\mathcal{G}} = \widehat{\mathcal{G}}^*$. For uniform scores $s(\cdot, q) = s_0 > 0$ and $\delta(\cdot, q) = \delta_0 > 0$, a plausible instantiation of $r$ would be:*
$$r(\widehat{\mathcal{G}}, q) := \frac{|\widehat{\mathcal{G}} \cap \widehat{\mathcal{G}}^*|s_0 - |\widehat{\mathcal{G}} \backslash \widehat{\mathcal{G}}^*|\delta_0}{|\widehat{\mathcal{G}}^*|s_0} \in [-\frac{\delta_0}{s_0}, 1]$$

However, due to the lack of the oracle set $\widehat{\mathcal{G}}^*$, existing methods resort to weak supervision signals $\widehat{\mathcal{G}}^w$ derived from some heuristics, such as query-answer (q-a) shortest paths (Zhang et al., 2022; Luo et al., 2023). While easy to extract, this approximation introduces a fundamental mismatch between $\widehat{\mathcal{G}}^*$ and $\widehat{\mathcal{G}}^w$, undermining retriever training in two key ways: **(I) Incompleteness** ($\widehat{\mathcal{G}}^* \setminus \widehat{\mathcal{G}}^w \neq \varnothing$). The shortest-path supervision often omits crucial reasoning components required to justify an answer. **(II)**

**Spurious inclusion ($\widehat{\mathcal{G}}^w \setminus \widehat{\mathcal{G}}^* \neq \varnothing$).** Conversely, shortest paths may include semantically irrelevant or misleading information. The examples of these two cases can be seen in Appendix E.1. Together, these issues underscore the need for more faithful supervision signals that better approximate $\widehat{\mathcal{G}}^*$.

### 3.2 COMPUTATIONAL CHALLENGE IN ITERATIVELY REFINED GRAPH-BASED RAG

To cope with the weak supervision, it is necessary to *refine the retrieved contents iteratively*. In other words, the retriever needs to maximize $r$ under a strict budget of LLM evaluations $C \ll |\mathcal{G}|$:

$$\max r(\widehat{\mathcal{G}}, q), \text{ s.t., } \widehat{\mathcal{G}} \in \{\widehat{\mathcal{G}}^{(i)}\}_{i=0}^T, T \leq C \quad (1)$$

where $\{\widehat{\mathcal{G}}^{(i)}\}_{i=0}^T$ is a sequence of subsets refined iteratively, initialized from $\widehat{\mathcal{G}}^{(0)} = \varnothing$ and updated via $\widehat{\mathcal{G}}^{(i+1)} = \text{ALG}\left(\left\{\widehat{\mathcal{G}}^{(j)}, r(\widehat{\mathcal{G}}^{(j)}, q)\right\}_{j=0}^i\right)$, and $\text{ALG}(\cdot)$ denotes a certain iterative graph-based RAG algorithm. However, as shown in the following proposition, solving for Eq. 1 is intractable.

**Proposition 3.1** *For any algorithm interacting with $r(\cdot, q)$ in Eq. 3.1 and $\mathcal{G}$, after $T$ rounds, achieving $\mathbb{P}\left(\exists i \in [T] : r\left(\widehat{\mathcal{G}}^{(i)}, q\right) = 1\right) \geq 1 - \varepsilon$ with $\varepsilon \in (0, 1)$ requires $T \geq \Omega\left(\frac{(1-\varepsilon)N}{\log N}\right)$. Additionally, if $|\widehat{\mathcal{G}}^{(i)}|, i \in [T]$ is fixed as a constant, then $T \geq \Omega((1-\varepsilon)N)$.*

See Appendix B.1 for the proof. In practice, since $N$ is large, identifying the $\widehat{\mathcal{G}}^*$ is essentially computationally intractable, which motivates us to consider more cost-efficient iterative approaches.

### 3.3 REPRESENTATION CHALLENGE IN RETRIEVED KNOWLEDGE

Beyond the two key properties of $r(\cdot, q)$ in Definition 3.1, LLMs are also sensitive not only to *what* facts are retrieved but also to *how* they are presented. Specifically, empirical studies indicate that LLMs perform best with logically coherent and contextually organized fact sequences (Chen et al., 2024c; Guo et al., 2025c), but suffer from disordered inputs due to the inherent position bias and reasoning constraints (Xiao et al., 2023; Jin et al., 2024; Yang et al., 2025). However, graph-based retrievers are typically designed to be permutation-invariant for $\tau \in \mathcal{G}$ and output unstructured sets of retrieval units (e.g., triples (Li et al., 2024a) or entities (Mavromatis & Karypis, 2024)). As LLMs exhibit strong sensitivity to structure and ordering, consequently, even a retrieved set $\widehat{\mathcal{G}}$ with high coverage of $\widehat{\mathcal{G}}^*$ may yield *underspecified rewards* if the retrieved contents are poorly organized or misaligned with the LLM's reasoning preferences. Misaligned representation of retrieved information even exacerbates the complexity of identifying $\widehat{\mathcal{G}}^*$ in the established optimization problem, and motivates the need for structure-aware alignment in graph-based RAG.

## 4 WEAK-TO-STRONG GRAPH-BASED RETRIEVER ALIGNMENT

To align the weak retrievers to LLMs, we present `Refined graph-based RAG`(ReG), which refines the weak supervision and aligns the retrieved knowledge to the favorite forms of LLMs.

### 4.1 REFINING SUPERVISION SIGNALS WITH LLMS

To bridge the gap between $\widehat{\mathcal{G}}^*$ and $\widehat{\mathcal{G}}^w$ in a cost-efficient way, first constructs a candidate pool $\mathcal{P}$, to cover diverse reasoning patterns, and then leverages LLMs to identify high-quality supervision signals from the candidates.

**Multi-Faceted Candidate Generation.** Since shortest paths provide high-recall coverage (Li et al., 2024a), we construct $\mathcal{P}$ based on $\mathcal{P}_{sp}$, where each $P \in \mathcal{P}_{sp}$ denotes a $q$-$a$ shortest path. To capture reasoning signals that shortest paths may overlook (Gu et al., 2021), we incorporate two auxiliary subsets:

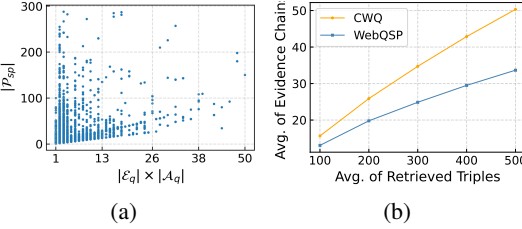

(a)     (b)

Figure 2: Scaling trends of: (a) $\mathcal{P}_{sp}$ size versus the number of $q$-$a$ pairs, and (b) the number of generated reasoning chains versus retrieved triples.

- $\mathcal{P}_q$: *Query-centric neighborhoods*, *i.e.*, one-hop neighbors around each query entity, enabling LLMs to incorporate the direct properties of the query entities into their reasoning.
- $\mathcal{P}_a$: *Answer-centric neighborhoods*, which enable comparisons across answer candidates based on numeric or categorical attributes.

Essentially, each candidate in $\mathcal{P}_q$ and $\mathcal{P}_a$ can be considered as a single-step reasoning path. The final candidate pool is: $\mathcal{P} := \mathcal{P}_{sp} \cup \mathcal{P}_a \cup \mathcal{P}_a$, providing a multi-faceted pool for LLM-guided refinement.

As shown in Fig. 2a, $\mathcal{P}_{sp}$ can grow superlinearly with the number of $q$-$a$ pairs due to the redundancy in the candidate pool, which brings additional computational overhead. To optimize efficiency, we compress $\mathcal{P}$ via structural merging to reduce redundancy. Details are given in Appendix E.2.

**LLM-Guided Candidate Refinement.** We leverage an LLM to identify plausible reasoning chains from the candidate pool $\mathcal{P}$. Each candidate path $P \in \mathcal{P}$ is textualized as a directed chain-of-entities that preserves logical flow for LLM reasoning. Using in-context learning (ICL) (Brown et al., 2020) with explanation-based demonstrations, we prompt the LLM to identify a subset of candidates $\widehat{\mathcal{P}}^+ \subseteq \mathcal{P}$ that provides sufficient and logically coherent evidence for answering the query. The detailed prompt can be found in Appendix G. Then, we extract the triples from $\widehat{\mathcal{P}}^+$ as refined supervision signals $\widehat{\mathcal{G}}^+$.

**Theoretical Discussion.** Essentially, the LLM-refined supervision signals $\widehat{\mathcal{G}}^+$ provide a more accurate and semantically aligned approximation to $\widehat{\mathcal{G}}^*$ than weak heuristics $\widehat{\mathcal{G}}^w$. Empirically, as validated in experiments (Sec. 5), the refined supervision demonstrates higher quality when applied to stronger training targets for the retriever.

Table 1: Compression Effectiveness of the complexity control step.

| GrailQA Dataset Gu et al. (2021) | # Avg | # Max |
|---|---|---|
| Raw Candidates | 275.16 | 98602 |
| + Complexity Control | 13.54 | 137 |
| **Compression Ratio** | ~5% | ~0.14% |

Meanwhile, ReG can overcome the computational bottleneck of the black-box optimization: as shown in Table 1, our structural merging yields substantial compression (down to 5% of the original candidate size on average), which is tractable for LLMs with standard reasoning capabilities under a constant order complexity $\mathcal{O}(1)$, despite the large $|\mathcal{G}|$.

**Retriever Training.** With the refined supervision, we can boost the retriever across different architectures (*e.g.*, MLP, GNNs, or LLMs) and retrieval units (*e.g.*, triple, entity, or path):

$$\max_{\theta} \mathbb{E}_{(q,\mathcal{A},\mathcal{G})\sim\mathcal{D}} \left[ \mathbb{P}_{\theta}(\widehat{\mathcal{G}}^+ \mid q, \mathcal{G}) \right], \text{ where } \widehat{\mathcal{G}}^+ := \texttt{SignalRefiner}(q, \mathcal{A}_q, \mathcal{G}, \texttt{LLM}) \quad (2)$$

where $\mathbb{P}_{\theta}$ denotes the retriever distribution parameterized by $\theta$, $\texttt{SignalRefiner}$ represents LLM-based refiner. See Appendix E.3 for instantiations of Eq. 2 across various retrieval granularities.

## 4.2 STRUCTURE-AWARE POST-RETRIEVAL RE-ORGANIZATION

To bridge the representation gap between retrieval and reasoning (Sec. 3.3), ReG also transforms the retrieved triples $\widehat{\mathcal{G}}$ into coherent evidence chains. Essentially, as the KG is originally organized in a logically coherent form, we align the retrieved results following the order in the KG.

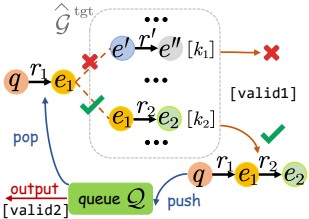

Figure 3: Illustration of BFS-guided chain expansion.

**Chain Expansion.** Specifically, we perform BFS-guided chain expansion over $\widehat{\mathcal{G}}$, which starts from query-anchored triples, defined as $\widehat{\mathcal{G}}^{\text{src}} := \{(h, r, t) \in \widehat{\mathcal{G}} \mid h \in \mathcal{E}_q \vee t \in \mathcal{E}_q\}$, and traverses through the left subgraph $\widehat{\mathcal{G}}^{\text{tgt}} := \widehat{\mathcal{G}} \setminus \widehat{\mathcal{G}}^{\text{src}}$. Each chain is incrementally extended by appending entity-linked triples, where a new triple $(h_{k+1}, r_{k+1}, t_{k+1})$ can be appended only if its head entity $h_{k+1}$ matches the tail $t_k$ of the last triple in the current chain. Alternatively, one could also apply other expansion methods, such as beam search. The chain expansion can be applied to various input units (*e.g.*, triple and entity levels). More details, including the complexity discussions of path expansion, can be found in Appendix E.4.

## 5 EXPERIMENTS

### 5.1 EXPERIMENTAL SETUP

We conduct comprehensive empirical studies to examine the effectiveness, efficiency, generalizability, and transferability of ReG in addressing the challenges in graph-based RAG (Sec. 3). Specifically, we aim to address the following research questions: (**RQ1**) How effective is ReG in handling complex multi-hop reasoning tasks across various retrieval methods? (**RQ2**) If ReG really improves the supervision quality, can ReG improve the data-efficiency? (**RQ3**) What is the contribution of each design component to the overall performance? (**RQ4**) Can ReG also enhance state-of-the-art reasoning LLMs? (**RQ5**) Are LLM-refined signals transferable across different backbone LLMs? (**RQ6**) Do LLM-refined signals enable better out-of-distribution generalization to zero-shot settings?

**Datasets.** We adopt four prominent and challenging KGQA datasets that necessitate multi-hop reasoning – WebQSP (Yih et al., 2016), CWQ (Talmor & Berant, 2018), and GrailQA (Gu et al., 2021). All datasets utilize Freebase (Bollacker et al., 2008) as the underlying KG. For WebQSP and CWQ, we follow Li et al. (2024a) and establish WebQSP-sub and CWQ-sub, where we remove samples whose answer entities are *absent* from the KG, to better evaluate the capability of LLM reasoners to produce factually consistent answers grounded in the external knowledge sources. The details of the evaluated datasets are given in the Appendix D.1.

**Baselines.** We mainly compare against several widely used graph-based RAG methods, including UniKGQA (Jiang et al., 2022), KD-CoT (Wang et al., 2023a), EtD (Liu et al., 2024a), StructGPT (Jiang et al., 2023), ToG (Sun et al., 2023), RoG (Luo et al., 2023), G-Retriever (He et al., 2024), SubgraphRAG (Li et al., 2024a), GNN-RAG (Mavromatis & Karypis, 2024), and GraphRAG-FI (Guo et al., 2025b). More details are given in Appendix D.2.

**Metrics.** Besides the commonly reported Macro-F1 and Hit, we also include Micro-F1 to account for the imbalance in the number of answers across samples and Hit@1 for a more inclusive evaluation.

**Implementation Details.** (I) *Retrieval.* We primarily use GPT-4o-mini and LLaMA-3.1-8B as backbone LLMs for supervision refinement, and select the retriever that achieves the best validation performance for downstream steps. As ReG is agnostic to both retriever architectures and retrieval units, we instantiate it with three representative settings: (1) triple-level retrieval with an MLP-based retriever; (2) entity-level retrieval with a GNN-based retriever; and (3) path-level retrieval with an LLM-based retriever. We refer to these variants as ReG@Triple, ReG@Entity, and ReG@Path, respectively. (II) *Reasoning.* We perform zero-shot reasoning with different LLMs, including LLaMA-3-8B, GPT-4o-mini, GPT-4o, QwQ-32B, and DeepSeek-R1, with temperature set to 0 and random seed fixed to 42 for reproducibility. More details of the implementations are given in Appendix F.

Table 2: Question-answering performance on WebQSP and CWQ. Best results are in bold.

| | WebQSP | | CWQ | |
|---|---|---|---|---|
| | Macro-F1 | Hit | Macro-F1 | Hit |
| UniKGQA | 72.2 | - | 49.0 | - |
| KD-CoT | 52.5 | 68.6 | - | 55.7 |
| ToG (GPT-4) | - | 82.6 | - | 67.6 |
| StructGPT | - | 74.69 | - | - |
| G-Retriever | 53.41 | 73.46 | - | - |
| RoG | 70.26 | 86.67 | 54.63 | 61.94 |
| EtD | - | 82.5 | - | 62.0 |
| GNN-RAG | 71.3 | 85.7 | 59.4 | 66.8 |
| GraphRAG-FI | 75.98 | 91.89 | 60.34 | 71.12 |
| SubgraphRAG (Llama3.1-8B) | 70.57 | 86.61 | 47.16 | 56.98 |
| SubgraphRAG (GPT-4o-mini) | 77.67 | 91.22 | 55.41 | 64.97 |
| SubgraphRAG (GPT-4o) | 78.24 | 90.91 | 59.42 | 67.49 |
| **ReG@*Triple*** | | | | |
| ReG (Llama3.1-8B) | 69.91 | 87.39 | 51.24 | 65.02 |
| ReG (GPT-4o-mini) | 77.98 | **92.87** | 60.55 | 67.66 |
| ReG (GPT-4o) | **78.76** | 91.4 | 62.34 | **71.51** |
| **ReG@*Entity*** | | | | |
| ReG (Llama3.1-8B) | 65.59 | 86.0 | 49.73 | 61.8 |
| ReG (GPT-4o-mini) | 76.78 | 92.26 | 58.13 | 66.24 |
| ReG (GPT-4o) | 77.88 | 90.97 | **62.58** | 68.91 |
| **ReG@*Path*** | | | | |
| ReG (Llama3.1-8B) | 70.88 | 84.71 | 49.58 | 59.9 |
| ReG (GPT-4o-mini) | 77.18 | 89.8 | 56.58 | 64.84 |
| ReG (GPT-4o) | 78.4 | 89.56 | 60.79 | 67.23 |

### 5.2 MAIN RESULTS ON WEBQSP AND CWQ

**Overall Performance (RQ1).** Tables 2 and 3 report evaluation results for ReG across three retrieval levels, *i.e.*, Triple, Entity, and Path, paired with downstream reasoners of varying strength, including LLaMA-3-8B, GPT-4o-mini, and GPT-4o. ReG achieves state-of-the-art (SOTA) performance across all four datasets, with all 12 metrics outperforming existing baselines when used with GPT-4o-mini or more advanced GPT-4o. Notably, ReG requires no finetuning of LLMs and invokes only one single reasoning call per query. Moreover, different retrieval levels of ReG excel on different metrics and datasets, reflecting its flexibility in accommodating varied reasoning demands and evaluation goals. For instance, the Path-level variant improves Micro-F1 by 5.6% on WebQSP-Sub, while both the Triple and Entity levels perform strongly on CWQ (*e.g.*, ↑10.1% Micro-F1 on CWQ-Sub). Also, ReG scales gracefully with LLM capabilities, maintaining consistent performance gains from LLaMA-3-8B to GPT-4o, especially for

Table 3: Question-answering performance on WebQSP-sub and CWQ-sub. Best results are in bold. Avg. Rank reports the average rank of the evaluated methods across the four evaluation metrics.

| | Webqsp-Sub | | | | | CWQ-sub | | | | |
|---|---|---|---|---|---|---|---|---|---|---|
| | Macro-F1 | Micro-F1 | Hit | Hit@1 | Avg.Rank | Macro-F1 | Micro-F1 | Hit | Hit@1 | Avg.Rank |
| G-Retriever | 54.13 | 23.84 | 74.52 | 67.56 | 11 | - | - | - | - | - |
| RoG-Joint | 72.01 | 47.7 | 88.9 | 82.62 | 9 | 58.61 | 52.12 | 66.22 | 61.17 | 9.25 |
| RoG-Sep | 67.94 | 43.1 | 84.03 | 77.61 | 10 | 57.69 | 52.83 | 64.64 | 60.64 | 9.75 |
| SubgraphRAG (GPT-4o-mini) | 78.46 | 57.08 | 92.43 | 88.01 | 5.25 | 62.18 | 56.86 | 72.82 | 66.57 | 8 |
| SubgraphRAG (GPT-4o) | 79.4 | 58.91 | 92.43 | 87.75 | 4.25 | 66.48 | 61.3 | 75.14 | 69.42 | 6.25 |
| ReG@Triple (GPT-4o-mini) | 78.91 | 59.43 | **94.36** | 88.2 | 3 | 67.99 | 64.91 | 75.53 | 71.38 | 3.75 |
| ReG@Triple (GPT-4o) | **80.08** | 57.88 | 93.14 | **88.97** | 2.5 | 68.91 | **67.5** | **77.81** | 72.23 | 1.5 |
| ReG@Entity (GPT-4o-mini) | 77.84 | 57.03 | 93.78 | 86.79 | 5.75 | 66.74 | 62.63 | 75.95 | 70.65 | 4.5 |
| ReG@Entity (GPT-4o) | 79.03 | 56.92 | 92.5 | 88.77 | 4.5 | **69.62** | 66.35 | 76.16 | **73.14** | 1.5 |
| ReG@Path (GPT-4o-mini) | 78.44 | 60.55 | 91.4 | 85.76 | 6 | 65.01 | 61.77 | 73.24 | 66.92 | 6.75 |
| ReG@Path (GPT-4o) | 79.48 | **62.23** | 90.96 | 86.14 | 4.5 | 68.4 | 65.99 | 75.28 | 71.07 | 3.75 |

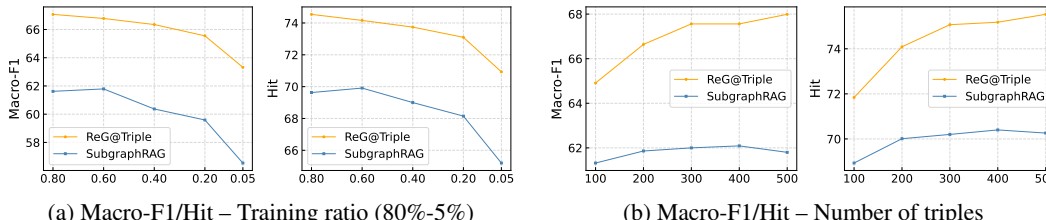

(a) Macro-F1/Hit – Training ratio (80%-5%)  (b) Macro-F1/Hit – Number of triples

Figure 4: Performance comparison (Macro-F1 and Hit) between ReG@Triple and SubgraphRAG on CWQ-Sub: (a) versus training set ratio and (b) versus number of retrieved triples. Evaluated using GPT-4o-mini with identical retriever/reasoner configurations for fair comparison.

CWQ & CWQ-Sub datasets which requires deeper reasoning hops and handling multiple reasoning chains from different query entities.

**Data Efficiency in Low-Resource Settings (RQ2).** To better verify that ReG indeed refines the weak supervision, we examine the performance of under the low-resource setting, i.e., only a small fraction of the full training dataset is available. As shown in Fig. 4a, we vary the proportion of training data from 80% down to 5%, comparing ReG@Triple against SubgraphRAG (Li et al., 2024a), which is a state-of-the-art Triple-level baseline. We observe that ReG with only 5% training data achieves superior performance than the baseline counterpart with 80% training data. Besides, SubgraphRAG suffers more performance degradation as training data decreases, whereas ReG maintains more stable and stronger results. The robustness of ReG stems from the high-quality supervision signals refined via LLMs, which get rid of noises and spurious signals and provide better guidance to the retriever.

Moreover, since ReG requires LLM calls on only a small subset of the training dataset to yield a retriever that matches or outperforms fully supervised baselines, the overall computational cost associated with LLM refinement remains low. This underscores the data efficiency of our method and makes it practical for real-world deployment under limited supervision and computational budgets.

**Scalability with Retrieval Volume.** It has been observed that the RAG performance may plateau or even degrade as retrieval volume increases (Li et al., 2024a), due to the adverse effects of irrelevant content (Wu et al., 2024) and the "lost-in-the-middle" phenomenon (Liu et al., 2024c). As shown in Fig. 4b, SubgraphRAG exhibits limited ability to benefit from additional retrieval content, even when equipped with GPT4o-mini. In contrast, ReG@Triple not only consistently outperforms SubgraphRAG but also shows more substantial performance gains as retrieval size increases, suggesting ReG retrieved more accurate and semantically aligned information for LLMs.

## 5.3 ABLATION STUDY (RQ3)

**Effects of Individual Modeules**. We ablate the two key components of ReG: **S(I)** – LLM-refined supervision signals, and **S(II)** – structure-aware reorganization. Specifically, we consider two variants: one merely removes **S(I)**; and the other removes both **S(I)** and **S(II)** to assess their individual and joint impacts across the three evaluated retrieval levels.

Table 4: Ablation studies across triple, entity, and path levels on CWQ-Sub, under GPT4o-mini inference. S(I) and S(II) represent the two stages introduced in Sec. 4.1 and 4.2, respectively.

| | Triple | | | | Entity | | | | Path | | | |
|---|---|---|---|---|---|---|---|---|---|---|---|---|
| | Macro-F1 | Micro-F1 | Hit | Hit@1 | Macro-F1 | Micro-F1 | Hit | Hit@1 | Macro-F1 | Micro-F1 | Hit | Hit@1 |
| w/o S(I) & S(II) | 61.8 | 58.51 | 70.26 | 65.03 | 55.75 | 45.42 | 70.26 | 61.73 | - | - | - | - |
| w/o S(I) | 65.95 | 60.38 | 76.11 | 70.05 | 61.12 | 58.99 | 69.84 | 63.73 | 59.81 | 58.06 | 67.73 | 61.45 |
| ReG | 67.99 | 64.91 | 75.53 | 71.38 | 66.74 | 62.63 | 75.95 | 70.65 | 65.01 | 61.77 | 73.24 | 66.92 |

Table 5: QA performance of ReG@Triple under QwQ-32B inference. See additional results on DeepSeek-R1 in Appendix H.

| | WebQSP-Sub | | | CWQ-Sub | | | GrailQA | | |
|---|---|---|---|---|---|---|---|---|---|
| | Macro-F1 | Hit | Avg.Tokens | Macro-F1 | Hit | Avg.Tokens | Macro-F1 | Hit | Avg.Tokens |
| w/o S(I) & S(II) | 77.65 | 95.1 | 923.03 | 65.41 | 81.48 | 1351.14 | 78.73 | 89.85 | 886.13 |
| w/o S(I) | 76.51 | 94.55 | 702.82 | 66.38 | 80.2 | 971.2 | 81.26 | 90.3 | 710.4 |
| ReG | 77.58 | 94.52 | 612.51 | 68.6 | 82.02 | 923.63 | 82.19 | 91.31 | 704.9 |

As shown in Table 4, removing either component consistently degrades performance across all settings. The full model, which combines **S(I)** and **S(II)** achieves the best results, highlighting the complementary roles of **S(I)** and **S(II)**: LLM-refined supervision improves retriever training quality, while structure-aware reorganization enhances the logical coherence of retrieved evidence, aligned to our discussion in Sec. 3 on the benefits of tackling both challenges.

**Multi-Hop & Multi-Entity QA**. Fig. 5 reports ablation results on two challenging question types: multi-hop questions, where answers lie multiple hops away from the query entities, and multi-entity questions, which require reasoning over multiple query anchors. ReG shows notable gains on both types, indicating its strength in handling deep and compositional reasoning. The LLM-refined supervision proves especially effective for entity- and path-level retrieval, while structure-aware reorganization alone yields substantial improvements, particularly in multi-entity settings where aligning disparate reasoning chains is critical.

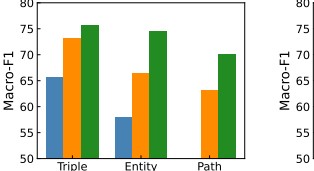 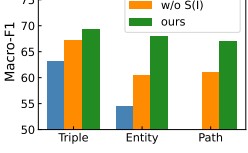

(a) Multi-Entity (53.31%)    (b) Multi-Hop (71.55%)

Figure 5: Ablations across three retrieval levels over multi-hop and multi-entity queries on the CWQ-Sub dataset. Note that there are some overlapped samples between the two query types. ReG consistently brings improvements with mutually beneficial components **S(I)** and **S(II)**.

## 5.4 EFFECTIVENESS ON STATE-OF-THE-ART LARGE REASONING MODELS (RQ4)

Following the advancement of LLMs, large reasoning models (LRMs) have gained huge success with significantly better capabilities in reasoning over the context (Guo et al., 2025a). To better understand the capabilities of LRMs in multi-hop reasoning and the effectiveness of ReG when paired with state-of-the-art large reasoning models, we apply ReG to QwQ-32B and DeepSeek-R1.

**Supervision Quality still Matters for LRMs.** As shown in Table 5, when paired with LRMs, retrievers trained with LLM-refined supervision signals consistently improve the performance by a noticeable margin. The improvements with more accurate and aligned retrievers reaffirm that retrieval quality remains a crucial factor for highly capable reasoning LLMs.

**ReG Improves Reasoning Efficiency.** LRMs often suffer from the *overthinking problem* (Chen et al., 2024b; Sui et al., 2025), which produces unnecessarily verbose reasoning traces with limited performance gains, ultimately degrading inference efficiency. To assess whether ReG alleviates this issue, we further examine the average reasoning tokens (Avg.Tokens) under different ablations.

As shown in Table 5, applying our structure-aware reorganization module leads to a substantial reduction in output length across all datasets. When further combined with refined supervision signals, the number of reasoning tokens can be reduced even further, accompanied by performance gains on various metrics, especially for datasets which require complex reasoning patterns. Together, these

Table 7: Retriever generalizability to out-of-distribution (OOD) KGs and reasoning patterns. $A \rightarrow B$ denotes training retrievers on the training set $A$ and evaluating on dataset $B$.

| | GrailQA → GrailQA-ZeroShot | | | | CWQ → GrailQA-ZeroShot | | | |
|---|---|---|---|---|---|---|---|---|
| | Macro-F1 | Micro-F1 | Hit | Hit@1 | Macro-F1 | Micro-F1 | Hit | Hit@1 |
| ReG w/o S(I) & S(II) | 75.59 | 44.02 | 84.02 | 81.67 | 67.29 | 35.16 | 76.1 | 73.79 |
| ReG w/o S(I) | 80.73 | 50.13 | 87.06 | 84.35 | 71.07 | 40.74 | 78.48 | 75.88 |
| ReG | 82.26 | 51.51 | 88.34 | 86.25 | 75.15 | 45.01 | 81.63 | 79.58 |

results suggest that ReG can improve both reasoning efficiency and quality, even for strong reasoning LLMs, indicating its potential as a broadly applicable enhancement to existing QA pipelines.

## 5.5 TRANSFERABILITY ANALYSIS OF LLM-REFINED SIGNALS (RQ5)

When refining weak supervision signals, different LLMs may introduce distinct biases. To study the robustness and transferability of the refined signals, we ablate signals refined by GPT-4o-mini, LLaMA-3–8B, their intersection, and their union, and evaluate QA performance via GPT-4o-mini.

As shown in Table 6, we observe that the refined signals by different LLMs generically bring improvements compared to the original weak supervision $\widehat{\mathcal{G}}^w$. Besides, the overall benefit of LLM refinement is more pronounced on CWQ than WebQSP. It is because of the more complex and compositional reasoning patterns in CWQ, which introduce more noise into weak heuristics and increase the need for effective signal filtering. Interestingly, neither the union nor the intersection of signals yields superior performance, indicating the need for more sophisticated strategies of collaborative refinement in the future.

Table 6: Question-answering performance on WebQSP and CWQ when training retrievers over supervision signals refined by different LLMs, denoted as $\widehat{\mathcal{G}}^+_{\text{gpt}}$ and $\widehat{\mathcal{G}}^+_{\text{llama}}$.

| ReG@Triple | WebQSP | | CWQ | |
|---|---|---|---|---|
| | Macro-F1 | Hit | Macro-F1 | Hit |
| $\widehat{\mathcal{G}}^w$ | 76.62 | 92.14 | 57.55 | 64.91 |
| $\widehat{\mathcal{G}}^+_{\text{gpt}}$ | 77.03 | 92.32 | **60.55** | **67.66** |
| $\widehat{\mathcal{G}}^+_{\text{llama}}$ | **77.98** | **92.87** | 60.11 | 67.35 |
| $\widehat{\mathcal{G}}^+_{\text{gpt}} \cap \widehat{\mathcal{G}}^+_{\text{llama}}$ | 77.51 | 92.75 | 59.38 | 66.47 |
| $\widehat{\mathcal{G}}^+_{\text{gpt}} \cup \widehat{\mathcal{G}}^+_{\text{llama}}$ | 77.03 | 92.63 | 60.19 | 67.53 |

## 5.6 OUT-OF-DISTRIBUTION GENERALIZATION IN ZERO-SHOT SETTINGS (RQ6)

We evaluate the out-of-distribution (OOD) generalization ability of retrievers trained with ReG. Specifically, we adopt the GrailQA dataset, which is explicitly designed to assess generalization to unseen schema items and domains. We first conduct few-shot training on the GrailQA-Train set and evaluate on GrailQA-Dev-Zeroshot subset. We then assess cross-dataset generalization by training on 20% of CWQ-Train data and testing on the zero-shot subset.

Shown as in Table 7, across both settings, retrievers trained with refined supervision consistently outperform those trained with weak proxy signals. These results suggest that LLM-refined supervision produces retrievers with stronger OOD generalization capabilities. This aligns with our earlier analysis in Sec. 3.1, which highlights two key limitations of heuristic-based supervision: (1) it often includes spurious paths which reach the correct answer but cannot reflect the actual reasoning logic required to answer the query, and (2) it may omit critical supporting information that explains why an answer is correct. In contrast, ReG results in more semantically faithful supervision and therefore yields stronger zero-shot generalization of retrievers.

## 6 CONCLUSIONS

In this work, we systematically studied the drawbacks of the weak retrievers in graph-based RAG. We showed that graph-based RAG can be formulated as a black-box combinatorial optimization problem. While resolving the original problem is computationally expensive due to the weak supervision and misaligned presentation of the retrieved results, we introduced ReG to align the weak retrievers to LLMs. ReG first refines weak supervision signals from diverse candidate subgraphs guided by LLMs, and then structurally reorganizes retrieval outputs to better match the preferences of LLM reasoning. Extensive experiments across datasets, retriever types, and reasoning LLMs demonstrated that ReG consistently improves both retrieval accuracy, data efficiency, reasoning efficiency, and OOD generalizability, even when paired with state-of-the-art reasoning LLMs.

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

## LLM USAGE DISCLOSURE

In our work, we mainly use LLMs to polish the paper writing and improve the clarity. Meanwhile, from the research perspective, we study how to improve the reasoning and mitigate the hallucination of LLMs.

## REPRODUCIBILITY STATEMENT

We describe our dataset details in Appendix D.1. For additional training details, see Appendix F. For prompt templates, see Appendix G. With the chairs' approval, we will also provide an anonymous code link during the rebuttal period.

## A    NOTATION SUMMARY

Table 8: Notations used in this work.

| Symbol | Description |
|---|---|
| $\mathcal{G} := \{(h, r, t) \mid h, t \in \mathcal{E}, r \in \mathcal{R}\}$ | Original knowledge graph (KG), a set of triples $(h, r, t)$ |
| $\mathcal{E}_q$ | Set of query entities extracted from question $q$ |
| $\mathcal{A}_q$ | Set of answer entities associated with question $q$ |
| $P := (\tau_1, \dots, \tau_L)$ | Length-$L$ reasoning path where each $\tau_i = (h_i, r_i, t_i) \in \mathcal{G}$ |
| $\mathbf{r}_P = (r_1, \dots, r_L)$ | Relation path corresponding to reasoning path $P$ |
| $\widehat{\mathcal{G}}$ | Retrieved subgraph from the KG (used for inference) |
| $\widehat{\mathcal{G}}^{\text{src}} := \{(h, r, t) \in \widehat{\mathcal{G}} \mid h \in \mathcal{E}_q \vee t \in \mathcal{E}_q\}$ | Query-anchored triples in the retrieved subgraph $\widehat{\mathcal{G}}$ |
| $\widehat{\mathcal{G}}^w$ | Weak supervision signals, *i.e.*, a noisy subgraph used for training |
| $\widehat{\mathcal{G}}^*$ | Oracle subgraph (ideal set of supporting facts for answering $q$) |
| $\mathcal{P} := \{P_i\}_{i=1}^{|\mathcal{P}|}$ | Candidate path pool used for refinement |
| $\widehat{\mathcal{P}}^+$ | LLM-refined candidate paths selected from $\mathcal{P}$ |
| $\widehat{\mathcal{G}}^+$ | LLM-refined supervision subgraph derived from $\widehat{\mathcal{P}}^+$ |

## B    COMPLEMENTARY DETAILS OF FORMULATIONS

### B.1    PROOF OF PROPOSITION 3.1

*proof.* Let $\widehat{\mathcal{G}}^* \in \mathcal{H}$ denotes the ground-truth hypothesis, where the hypothesis space satisfies $|\mathcal{H}| = 2^N$. Let $r^{(1:T)} := (r^{(1)}, \cdots, r^{(T)})$ denote the sequence of observed responses collected over $T$ rounds of interaction. From Fano's inequality,

$$H(\widehat{\mathcal{G}}^* | r^{(1:T)}) \leq \mathcal{O}(1) + \varepsilon \log |\mathcal{H}| \tag{3}$$

From definitions of mutual information,

$$I(\widehat{\mathcal{G}}^*; r^{(1:T)}) = H(\widehat{\mathcal{G}}^*) - H(\widehat{\mathcal{G}}^* | r^{(1:T)}), \tag{4}$$

where $H(\widehat{\mathcal{G}}^*) = \log |\mathcal{H}|$, then we have,

$$I(\widehat{\mathcal{G}}^*; r^{(1:T)}) \geq (1 - \varepsilon) \log |\mathcal{H}| - \mathcal{O}(1) \tag{5}$$

From chain rules of mutual information,

$$I(\widehat{\mathcal{G}}^*; r^{(1:T)}) = \sum_{t=1}^{T} I(\widehat{\mathcal{G}}^*; r^{(t)} | r^{(<t)}) \tag{6}$$

Recall Eq. 3.1, since the $r(\cdot, q)$ depends on the number of correctly selected ($\sim \mathcal{O}(1)$) and incorrectly selected ($\sim \mathcal{O}(N)$) items, the total number of possible reward values is at most $\mathcal{O}(N)$. Therefore, for the $t$-th term, we have,

$$I(\widehat{\mathcal{G}}^*; r^{(t)}|r^{(<t)}) \leq H(r^{(t)}|r^{(<t)}) \leq H(r^{(t)}) \leq \log|\mathcal{R}| \leq \mathcal{O}(\log N), \tag{7}$$

where $\mathcal{R} := \{r(\mathcal{S}, q) \mid \mathcal{S} \subseteq \mathcal{G}\}$. THe first inequality holds via the property of mutual information. Hence,

$$I(\widehat{\mathcal{G}}^*; r^{(1:T)}) \leq T\mathcal{O}(\log N) \tag{8}$$

From Eq. 8 and 5, we have,

$$T \geq \Omega\left(\frac{(1-\varepsilon)N}{\log N}\right) \tag{9}$$

If we fix the size $|\widehat{\mathcal{G}}^{(i)}|$ in each round $i \in [T]$, then $|\mathcal{R}| \leq |\widehat{\mathcal{G}}^*|$ and hence $|\mathcal{R}| \sim \mathcal{O}(1)$, and thus $T \geq \Omega((1-\varepsilon)N)$

## B.2 A SPECIFIC VERSION

Proposition 3.1 provides a general lower bound on the number of queries $T$ for any algorithm interacting with the black-box evaluator. We now analyze a specific iterative algorithm: Slightly different from Eq. 3.1, here we define the reward of a subset $\mathcal{S} \subset \mathcal{G}$ as,

$$r(\mathcal{S}, q) := \frac{|\mathcal{S} \cap \widehat{\mathcal{G}}^*|s_0 - |\mathcal{S}\backslash\widehat{\mathcal{G}}^*|\delta_0}{|\mathcal{S}|s_0} \in [-\frac{\delta_0}{s_0}, 1], \tag{10}$$

We define $r_0 := r(\mathcal{G}, q)$, $i.e.$, the reward on the universe set $\mathcal{G}$, and let $r_0 > 0$. In each round of evaluation $i \in \{0, \cdots, T-1\}$, a set $\mathcal{S}^{(i)}$ of size $S \ll N$ is drawn uniformly at random from $\mathcal{G}$ and returned to the candidate pool after the round. Let the identified set $\widehat{\mathcal{G}}^{(i)}$, initialized with $\widehat{\mathcal{G}}^{(0)} = \oslash$, evolves via the following rule:

$$\widehat{\mathcal{G}}^{(i+1)} \leftarrow \widehat{\mathcal{G}}^{(i)} \cup \mathcal{S}^{(i)}, \text{ if } r(\mathcal{S}^{(i)}, q) > \tau, \tag{11}$$

where the threshold $\tau \in (p, 1)$ is a constant. Proposition B.1 characterizes the sample complexity of recovering sparse high-relevance sets for the above algorithmic procedure.

**Proposition B.1** *Guided by $r(\cdot, q)$ in Eq. 10 and rule 11, after T rounds, achieving*

$$\mathbb{P}(\widehat{\mathcal{G}}^* \subset \widehat{\mathcal{G}}^{(T)}) \geq 1 - \varepsilon \tag{12}$$

*with $\varepsilon \in (0, 1)$ requires*

$$T = \Omega\left(e^{2\tau^2 S}\frac{N}{S}\log\frac{1}{\varepsilon}\right). \tag{13}$$

Specifically, stricter thresholds (higher $\tau$) exponentially increase the required rounds, as only subsets $\mathcal{S}^{(i)}$ with higher signal-to-noise ratio meet the acceptance criterion $r(\mathcal{S}^{(i)}, q) > \tau$, and thus guarantees improvement in the Jaccard similarity between the final identified set $\widehat{\mathcal{G}}^{(T)}$ and $\widehat{\mathcal{G}}^*$.

*proof.* We start by deriving $\mathbb{P}(t \notin \widehat{\mathcal{G}}^{(T)})$ for a specific oracle item $t \in \widehat{\mathcal{G}}^*$. We have,

$$\mathbb{P}(t \notin \widehat{\mathcal{G}}^{(T)}) = \prod_{i=0}^{T-1} \mathbb{P}(t \notin \widehat{\mathcal{G}}^{(i+1)}|t \notin \widehat{\mathcal{G}}^{(i)}) \tag{14}$$

Now we focus on $\mathbb{P}(t \notin \widehat{\mathcal{G}}^{(i+1)}|t \notin \widehat{\mathcal{G}}^{(i)})$ for $i \in \{0, \ldots, T-1\}$,

$$\mathbb{P}(t \notin \widehat{\mathcal{G}}^{(i+1)}|t \notin \widehat{\mathcal{G}}^{(i)}) \tag{15}$$

$$= \mathbb{P}(t \notin \widehat{\mathcal{G}}^{(i+1)}|t \notin \widehat{\mathcal{G}}^{(i)}, t \in \mathcal{S}^{(i)})\mathbb{P}(t \in \mathcal{S}^{(i)}) + \mathbb{P}(t \notin \widehat{\mathcal{G}}^{(i+1)}|t \notin \widehat{\mathcal{G}}^{(i)}, t \notin \mathcal{S}^{(i)})\mathbb{P}(t \notin \mathcal{S}^{(i)}) \tag{16}$$

For $i$-th round, a subset $\mathcal{S}^{(i)}$ of size $S$ is drawn at random from $\mathcal{G}$ without replacement. Therefore, we have $\mathbb{P}(t \in \mathcal{S}^{(i)}) = \frac{S}{N}$ and $\mathbb{P}(t \notin \mathcal{S}^{(i)}) = 1 - \frac{S}{N}$. Also, it's easy to see $\mathbb{P}(t \notin \widehat{\mathcal{G}}^{(i+1)}|t \notin \widehat{\mathcal{G}}^{(i)}, t \notin \mathcal{S}^{(i)}) = 1$, and $\mathbb{P}(t \notin \widehat{\mathcal{G}}^{(i+1)}|t \notin \widehat{\mathcal{G}}^{(i)}, t \in \mathcal{S}^{(i)}) = \mathbb{P}(r(\mathcal{S}^{(i)}, q) \leq \tau|t \in \mathcal{S}^{(i)})$.

Define $K := |\widehat{\mathcal{G}}^*|$. Let *r.v.* $K_S$ be $|\mathcal{S}^{(i)} \cap \widehat{\mathcal{G}}^*|$, we know $K_S \sim \text{Hypergeometric}(N, K, S)$. Via Eq. 10, we have $K_S s_0 - (S - K_S)\delta_0 \leq S\tau s_0$, and,

$$K_S \leq S\frac{\tau s_0 + \delta_0}{s_0 + \delta_0} \triangleq S\theta, \tag{17}$$

where $\theta := \frac{\tau s_0 + \delta_0}{s_0 + \delta_0}$. Therefore,

$$\mathbb{P}(r(\mathcal{S}^{(i)}, q) \leq \tau | t \in \mathcal{S}^{(i)}) = \mathbb{P}(K_S \leq S\theta | t \in \mathcal{S}^{(i)}) \leq \mathbb{P}(K_S \leq S\theta), \tag{18}$$

and thus, we have,

$$\mathbb{P}(t \notin \widehat{\mathcal{G}}^{(i+1)} | t \notin \widehat{\mathcal{G}}^{(i)}) \leq 1 - (1 - \mathbb{P}(K_S \leq S\theta))\frac{S}{N} \triangleq 1 - p_0\frac{S}{N}, \tag{19}$$

where $p_0 := \mathbb{P}(K_S \geq S\theta)$. Now, Eq. 19 $\rightarrow$ Eq. 14, we have,

$$\mathbb{P}(t \notin \widehat{\mathcal{G}}^{(T)}) \leq \left(1 - p_0\frac{S}{N}\right)^T \tag{20}$$

Applying the union bound over all $K$ target items gives

$$\mathbb{P}(\widehat{\mathcal{G}}^* \not\subset \widehat{\mathcal{G}}^{(T)}) = \mathbb{P}\left(\bigcup_{t \in \widehat{\mathcal{G}}^*} t \notin \widehat{\mathcal{G}}^{(T)}\right) \leq \sum_{t \in \widehat{\mathcal{G}}^*} \mathbb{P}(t \notin \widehat{\mathcal{G}}^{(T)}) \leq K\left(1 - p_0\frac{S}{N}\right)^T \leq K\exp\left\{-\frac{p_0 ST}{N}\right\} \tag{21}$$

To guarantee that this probability is at most $\varepsilon$, it suffices that

$$K\exp\left\{-\frac{p_0 ST}{N}\right\} \leq \varepsilon, \tag{22}$$

which is equivalent to

$$T \geq \frac{N}{p_0 S}\log\frac{K}{\varepsilon} \tag{23}$$

Now, let us again focus on $p_0$. Define $p := \mathbb{E}[K_S] = \frac{K}{N}$. Firstly, we solve $\theta - p$.

$$\theta - p = \frac{\tau s_0 + \delta_0}{s_0 + \delta_0} - p = \frac{\tau s_0 - (ps_0 - (1-p)\delta_0)}{s_0 + \delta_0} = \frac{\tau s_0 - r_0}{s_0 + \delta_0}, \tag{24}$$

which holds via the definition of $r_0 = r(\mathcal{G}, q) = \frac{Ks_0 - (N-K)\delta_0}{N} := ps_0 - (1-p)\delta_0$. Also, it's clear to see $r_0 < ps_0 < \tau s_0$ and thus $\theta > p$. Given that $r_0 > 0$, we have $\delta_0 < \frac{p}{1-p}s_0$. Therefore, we have,

$$\theta - p > \frac{\tau s_0 - ps_0}{s_0 + \frac{p}{1-p}s_0} = (1-p)(\tau - p) \tag{25}$$

Via Hoeffding-type concentration inequality for Hypergeometric$(N, K, S)$ tails, we have

$$p_0 = \mathbb{P}(K_S \geq S\theta) \leq \exp\left\{-2S(\theta - p)^2\right\}, \tag{26}$$

and thus,

$$T \geq e^{2S(\theta-p)^2}\frac{N}{S}\log\frac{K}{\varepsilon} \geq e^{2S(1-p)^2(\tau-p)^2}\frac{N}{S}\log\frac{K}{\varepsilon} \tag{27}$$

Considering $K \sim \mathcal{O}(1)$ and $p \sim \mathcal{O}(1/N)$, we have

$$T = \Omega\left(e^{2\tau^2 S}\frac{N}{S}\log\frac{1}{\varepsilon}\right) \tag{28}$$

## C    DETAILED DISCUSSION ON RELATED WORK

**Graph-based Retrieval.**    Prior work has explored retrieving from existing KGs like Wiki-Data (Vrandečić & Krötzsch, 2014) or Freebase (Bollacker et al., 2008), or augmenting text corpora with graph overlays (Edge et al., 2024; Gutiérrez et al., 2024; Guo et al., 2024) to improve relevance modeling during retrieval.  Due to the lack of an oracle in graph-based RAG, it often relies on heuristic priors or weak supervisions to guide the retrieval, which can be broadly categorized as: *Training-free* methods typically adopt graph-based heuristics (*e.g.,* personalized PageRank) (Gutiérrez et al., 2024; 2025) or LLM-guided stepwise exploration over the graph (Gu et al., 2022; Sun et al., 2023; Xiong et al., 2024). However, graph algorithms often underperform as they struggle to combine semantic and structural information (Luo et al., 2025b), while LLM-guided traversal suffers from high computational cost and is prone to local decision biases (Luo et al., 2025a). *Training-based* methods train parametric retrievers to recognize critical substructures (Li et al., 2024a; Mavromatis & Karypis, 2024; He et al., 2024). However, the training supervision is typically derived from noisy proxies such as query-answer shortest paths (Zhang et al., 2022; Luo et al., 2023), which may omit critical intermediate evidence and introduce spurious connections unrelated to the reasoning logic (Li et al., 2024a). Differently, we aim to bridge the gap by integrating the rich knowledge of LLMs to improve the quality of weak supervision for retriever training.

**RAG with LLM Feedback.** Recent efforts in text-based RAG have explored using LLM feedback to directly optimize the retriever capability. Shi et al. (2023) aligns retriever output distributions with the perplexity-reducing behavior of LMs. Li et al. (2024b) iteratively tunes the LLM generator and retriever by aligning data preferences between both modules. Han et al. (2025) trains an intermediary module by collecting preference data from both the LLM and the retriever to enhance RAG performance. While effective for document-level retrieval, in graph-based RAG, the combinatorial explosion of subgraph candidates and their intricate dependencies pose unique challenges.

**RAG with Post-retrieval Refinement.** In addition, to compensate for retrieval noise, prior work has also explored various post-hoc refinement strategies, including re-ranking retrieved contents (Jin et al., 2024), filtering irrelevant spans (Chen et al., 2025; Wang et al., 2023b; Guo et al., 2025b), or fine-tuning LLMs to tolerate noisy input (Yoran et al., 2023; Yu et al., 2024; Zhang et al., 2024; Jin et al., 2024). While these methods improve robustness, they largely assume the initial retrieval output to be fixed. Consequently, they hardly recover critical evidence missing from the initial retrieval stage, nor improve the capability of the upstream retriever itself.

## D    DETAILS OF DATASETS AND BASELINES

### D.1    DATASET DETAILS

**WebQSP** (Yih et al., 2016) is an enriched version of the WebQuestions dataset, containing 4,737 questions annotated with semantic parses. The questions require up to 2-hop reasoning over the KG.

**CWQ** (Talmor & Berant, 2018) builds on WebQSP by extending questions with additional constraints or entity chains to form more complex multi-hop queries, totaling 34,689 questions with reasoning depths up to 4 hops. Notably, over 50% of WebQSP test questions (or their variants) appearing in CWQ's training set, and vice versa (Li et al., 2024a).

**GrailQA** (Gu et al., 2021) is a large-scale dataset designed to evaluate generalization in KGQA across i.i.d., compositional, and zero-shot settings. It features a wide range of logical forms and multi-hop questions. We focus on the zero-shot subset, which requires reasoning over unseen schemas and relation compositions, making it ideal for assessing retriever generalization to out-of-distribution knowledge and reasoning patterns.

### D.2    BASELINE DETAILS

**UniKGQA** (Jiang et al., 2022) unifies retrieval and reasoning for multi-hop QA over the KGs by integrating both stages in model architecture and parameter learning, tightly relating the retrieval and reasoning processes.

**KD-CoT** (Wang et al., 2023a) enhances Chain-of-Thought reasoning in the LLMs for knowledge-intensive QA by verifying and refining intermediate steps through structured interaction with external knowledge, to reduce hallucinations and improve performance over standard CoT methods.

**EtD** (Liu et al., 2024a) is a two-stage framework for KGQA that combines lightweight GNN-based exploration with knowledge-enhanced prompting for LLM-based determination, enabling faithful reasoning over the KGs.

**StructGPT** (Jiang et al., 2023) introduces an Iterative Reading-then-Reasoning framework that equips the LLMs with specialized interfaces to iteratively gather evidence from structured data and perform focused reasoning.

**ToG** (Sun et al., 2023) introduces a training-free "LLM $\otimes$ KG" paradigm where an LLM agent iteratively explores related entities and relations on KGs via beam search to discover promising reasoning paths and perform traceable reasoning.

**RoG** (Luo et al., 2023) proposes a planning-retrieval-reasoning framework which first generates relation paths grounded by KGs as faithful plans. These plans are then used to retrieve valid reasoning paths from the KGs for LLMs to conduct faithful reasoning.

**G-Retriever** (He et al., 2024) perform RAG over textual graphs by formulating the task as a Prize-Collecting Steiner Tree optimization problem. It supports fine-tuning to enhance graph understanding via soft prompting.

**SubgraphRAG** (Li et al., 2024a) integrates a lightweight multilayer perceptron with a parallel triple-scoring mechanism for efficient and flexible subgraph retrieval while encoding directional structural distances to enhance retrieval effectiveness on capturing structural connections over KGs.

**GNN-RAG** (Mavromatis & Karypis, 2024) combines GNNs with the language capabilities of LLMs in a RAG framework by retrieving candidate answers via GNN-based KG retrieval and guiding LLM inference using verbalized KG reasoning paths.

**GraphRAG-FI** (Guo et al., 2025b) improves graph-based RAG by introducing a two-stage filtering mechanism and a logits-based selection strategy to reduce noise in retrieved information and balance external knowledge with LLMs' intrinsic reasoning.

# E METHODOLOGY DETAILS

## E.1 EXAMPLES OF THE GAP BETWEEN $\widehat{\mathcal{G}}^*$ AND $\widehat{\mathcal{G}}^w$

Existing retrievers are typically trained on query-answer ($q$-$a$) shortest paths as proxy supervisions (Zhang et al., 2022; Luo et al., 2023). While easy to extract, these proxy signals face two intrinsic limitations:

**(I) Spurious inclusion ($\widehat{\mathcal{G}}^w \setminus \widehat{\mathcal{G}}^* \neq \varnothing$).** Shortest paths may contain relations that are semantically irrelevant to the query demand. For instance, consider the question "Which countries border Spain", the path of "Spain $\xrightarrow{\text{currency-used}}$ Euro $\xrightarrow{\text{country-used}}$ Portugal" fails to capture the spatial relation of interest (*i.e.*, bordering), and thus provides misleading supervision.

**(II) Incompleteness ($\widehat{\mathcal{G}}^* \setminus \widehat{\mathcal{G}}^w \neq \varnothing$).** Some queries require additional structural information beyond $q$-$a$ shortest paths. For *aggregation*-type queries such as "how many distilled spirits are associated with tequilla", the correct answer is a numeric value derived from enumerating outgoing edges of the query entity. For *comparison*-type queries such as "Rocket engines with LOX oxidizer and chamber pressure ¡ 79.0", the $q$-$a$ shortest path "Liquid oxygen $\xrightarrow{\text{rocket\_engines}}$ Rocketdyne F-1" provides a reasonable connection, but lacks critical evidence for the numerical constraint, which is captured by the answer-centric neighborhoods: "Rocketdyne F-1 $\xrightarrow{\text{chamber\_pressure}}$ 70.0", enabling LLMs to perform necessary numerical filtering over candidate entity properties.

### E.2  PATH MERGING IN LLM-REFINEMENT

To preserve tractability and maintain minimal LLM usage overhead, we apply a structural merging step to compress the candidate path pool $\mathcal{P}$ into a compact set, while retaining structural diversity and logical coverage.

(I) **Answer Merging.** Only one representative answer $a^* \in \mathcal{A}_q$ is selected from $\mathcal{A}_q$, as correct answers typically share the same underlying reasoning path.

(II) **Relation-Chain Merging.** Paths sharing the identical relation path are merged into a unified candidate, removing redundancy without losing logical diversity. Here, the relation path of a path $P := (\tau_1, \ldots, \tau_L)$ is defined as the ordered sequence of relations $(r_1, \ldots, r_k)$ extracted from each triple $\tau_i = (h_i, r_i, t_i) \in P$ in the path.

### E.3  INSTANTIATION OF RETRIEVER-TRAINING OBJECTIVES

Eq. 2 gives a general-purpose MLE-based training objective which is model-agnostic and accommodates a wide range of retrieval units, including Triple, Entity, and Path. Here we give detailed instantiations of Eq. 2 across the three levels as follows.

**ReG@Triple**. Triples encapsulate atomic semantic relations between concepts and avoids the combinatorial explosion of subgraph enumeration. We formulate retriever training as binary classification over individual triples $\tau \in \mathcal{G}$, where each triple serves as a basic retrieval unit. A triple is labeled as positive if $\tau \in \widehat{\mathcal{G}}^+$, representing its semantic relevance to the query $q$. Each triple is encoded in context: its representation $z_\tau := z_\tau(\tau, \mathcal{G}, q)$ incorporates both the local graph neighborhood and the query embedding, enabling the retriever to make relevance judgments that are both structure-aware and query-aware. The retriever is optimized to maximize:

$$\max_{\theta} \mathbb{E}_{(q, \mathcal{A}_q, \mathcal{G}, \widehat{\mathcal{G}}^+) \sim \mathcal{D}} \left[ \sum_{\tau \in \mathcal{G}^+} \log p_\theta(\tau \mid z_\tau) + \sum_{\tau \in \mathcal{G} \backslash \mathcal{G}^+} \log(1 - p_\theta(\tau \mid z_\tau)) \right], \qquad (29)$$

where $p_\theta$ denotes the retriever, and $z_\tau := z_\tau(\tau, \mathcal{G}, q)$ denotes the contextualized representation of the triple $\tau$.

**ReG@Entity**. The entity-level retriever similarly treats individual entities $e \in \mathcal{E}$ as basic retrieval units. Entities that appear in $\widehat{\mathcal{G}}^+$ are labeled as relevant, forming the positive set $\widehat{\mathcal{E}}^+ := \mathcal{E}(\widehat{\mathcal{G}}^+)$. Each entity is encoded with a contextualized representation $z_e := z_e(e, \mathcal{G}, q)$ derived from its surrounding subgraph and the query. The model is trained to distinguish relevant entities from irrelevant ones via a binary cross-entropy loss:

$$\max_{\theta} \mathbb{E}_{(q, \mathcal{A}_q, \mathcal{G}, \widehat{\mathcal{G}}^+) \sim \mathcal{D}} \left[ \sum_{e \in \widehat{\mathcal{E}}^+} \log p_\theta(e \mid z_e) + \sum_{e \in \mathcal{E} \backslash \widehat{\mathcal{E}}^+} \log(1 - p_\theta(e \mid z_e)) \right]. \qquad (30)$$

**ReG@Path**. Following Luo et al. (2023), we fine-tune a language model to serve as a planning module that generates plausible relation paths given a query $q$. These relation paths act as high-level logical sketches, which describes the reasoning patterns needed to traverse the KG and arrive at the correct answer. Once generated, these paths are grounded to the KG by matching them against actual sequences of triples, yielding reasoning paths that lead to candidate answers. To train this planner, we use the relation paths extracted from each LLM-refined reasoning path $P \in \widehat{\mathcal{P}}^+$ as supervision signals. The training objective maximizes the likelihood of generating the correct relation path given the query:

$$\max_{\theta} \mathbb{E}_{(q, \mathcal{A}_q, \mathcal{G}, \widehat{\mathcal{P}}^+) \sim \mathcal{D}} \left[ \sum_{P \in \widehat{\mathcal{P}}^+} \log p_\theta^{\mathrm{LLM}}(\mathbf{r}(P) \mid q) \right], \qquad (31)$$

where $\mathbf{r}(P)$ denotes the relation path of the reasoning path $P$ (see definition in Sec. E.2).

---

**Algorithm 1** BFS-Based Reasoning Chain Expansion

---

**Require:** Retrieved triple set $\widehat{\mathcal{G}}$, query entities $\mathcal{E}_q$, max length $L$
**Ensure:** Set of reasoning chains $\mathcal{P}$
1: $\widehat{\mathcal{G}}^{\text{src}} \leftarrow \{(h, r, t) \in \widehat{\mathcal{G}} \mid h \in \mathcal{E}_q\}$        ▷ `Query-anchored triples`
2: $\widehat{\mathcal{G}}^{\text{tgt}} \leftarrow \widehat{\mathcal{G}} \setminus \widehat{\mathcal{G}}^{\text{src}}$        ▷ `Non-query triples`
3: $\mathcal{P} \leftarrow \emptyset$, $\mathcal{Q} \leftarrow \text{Queue}(\widehat{\mathcal{G}}^{\text{src}})$, $\mathcal{T}^{\text{vis}} \leftarrow \emptyset$      ▷ `Initialize paths and queue`
4: **while** $\mathcal{Q}$ is not empty **do**
5:     $P \leftarrow \mathcal{Q}.\text{dequeue}()$        ▷ `Current path` $(\tau_1, ..., \tau_k)$
6:     **if** $P \notin \mathcal{P}$ **then**
7:        $\mathcal{P} \leftarrow \mathcal{P} \cup \{P\}$
8:        $(h_k, r_k, t_k) \leftarrow \tau_k$        ▷ `Last triple in path`
9:        $(h_1, r_1, t_1) \leftarrow \tau_1$        ▷ `First triple in path`
10:     **end if**
11:     **if** $|P| \geq L$ **then continue**
12:     **end if**
13:     **for** each $\tau' = (h', r', t') \in \widehat{\mathcal{G}}^{\text{tgt}}$ **do**
14:        **if** $t_k == h'$ and $(h_1, t') \notin \mathcal{T}^{\text{vis}}$ **then**     ▷ `Head-tail matching`
15:           $P' \leftarrow P \oplus \tau'$        ▷ `Path extension`
16:           $\mathcal{Q}.\text{enqueue}(P')$
17:           $\mathcal{T}^{\text{vis}} \leftarrow \mathcal{T}^{\text{vis}} \cup \{(h_1, t')\}$     ▷ `Mark query-target pair` $(h_1, t')$ `as visited`
18:        **end if**
19:     **end for**
20: **end while**
21: **return** $\mathcal{P}$

---

### E.4 DETAILS OF TWO-STEP STRUCTURE-AWARE REORGANIZATION

#### E.4.1 BFS-BASED EVIDENCE CHAIN EXPANSION

We provide the full procedure of the BFS-based path expansion described in Sec. 4.2. Algorithm 1 expands each query-anchored triple $\tau \in \widehat{\mathcal{G}}^{\text{src}}$ by iteratively growing coherent paths through entity-matching across triples in $\widehat{\mathcal{G}}^{\text{tgt}}$. It's noteworthy that, for simplicity, the algorithm below only describes one expansion direction, *i.e.*, extending the chain left-to-right, such that the query entity always appears as the head entity of the first triple. In practice, we perform bidirectional expansion: we also consider chains where the query entity serves as the tail of the final triple, expanding right-to-left accordingly.

While alternative heuristics such as beam search exist, we prefer BFS as its length limit $L$ directly corresponds to the number of reasoning hops, offering a more interpretable and semantically grounded control than its less meaningful counterparts such as the beam width $k$ in the beam search.

#### E.4.2 STRUCTURE-AWARE MERGING

To further reduce redundancy and enhance semantic clarity, we merge structurally related paths using two operations:

† *Multi-answer merging.* Paths $P_1, \cdots, P_k$ are merged into a unified path if 1) rooted at the same query entity $e \in \mathcal{E}_q$, 2) sharing the same relation path $(r_1, \cdots, r_{|P_1|})$, but 3) ending in different target entities. This handles scenarios where a single reasoning logic yields multiple valid answers and reduce the complexity of reasoning paths.

† *Multi-entity merging.* After multi-answer merging, for multi-entity queries, *i.e.*, $|\mathcal{E}_q| > 1$, paths $P_1$ and $P_2$ with distinct sources $\text{src}(P_1) \neq \text{src}(P_2) \in \mathcal{E}_q$ but overlapping targets $\text{tgt}(P_1) \cap \text{tgt}(P_2) \neq \emptyset$ are merged as follows: 1) *Spatial Reordering.* $P_1$ and $P_2$ are placed consecutively in the evidence sequence to reinforce their logical coherence; 2) *Target Refinement.* The new shared target set is set as $\text{tgt}(P_1) \cap \text{tgt}(P_2)$. While we describe pairwise merging here, the operation naturally generalizes to $n$-entity queries.

For the scalability of the proposed pipeline, as shown in Fig. 2b, the number of generated chains grows approximately sub-linearly with the number of retrieved triples, indicating that our structure-aware re-organization strategy maintains tractable complexity even under larger retrieval budgets.

After the two steps, The query $q$ and reorganized retrieved evidence chains are integrated into a structured prompt template with in-context demonstrations (*c.f.*, Appendix G), guiding the LLM to generate factually grounded answers.

## F  DETAILS OF EXPERIMENTAL IMPLEMENTATION

We adopt `gte-large-en-v1.5` (Li et al., 2023b) as the frozen text encoder for query and triple representation. This 434M model achieves a strong trade-off between efficiency and retrieval quality on English corpora, as validated by the Massive Text Embedding Benchmark (MTEB) leaderboard (Muennighoff et al., 2022). In our main experiments, we evaluate three retrieval levels, `ReG@Triple`, `ReG@Entity` and `ReG@Path`. We introduce the detailed implementations as follows.

**ReG@Triple.** We adopt the objective defined in Eq. 29. For the retriever architecture, we follow Li et al. (2024a), leveraging Directional Distance Encoding (DDE) to capture structural relationships in the KG. The encoded features are passed to a lightweight MLP for binary classification over individual triples. We rank all triples $\tau \in \mathcal{G}$ by the retriever-output score $p_\theta(\tau)$ which quantifies its relevance to the query $q$, and the top-$K$ triples are selected to form the retrieved subgraph $\widehat{\mathcal{G}}$ for downstream LLM reasoning. We set $K = 500$ for GPT-4o and GPT-4o-mini, and $K = 200$ for LLaMA-3.1-8B, considering its relatively limited reasoning capacity.

**ReG@Entity.** We follow Eq. 30 as the training objective. The retriever is implemented as a GNN, specifically, a PNA (Principal Neighborhood Aggregation) network (Corso et al., 2020), which strengthens the expressive power of message passing by leveraging a rich combination of aggregators and degree-aware scalers. The retriever outputs a score $p_\theta(e)$ for each entity $e \in \mathcal{E}$. To maintain consistency with the `ReG@Triple` pipeline, we convert entity scores into triple scores by $s(\tau) := p_\theta(h) + p_\theta(t)$ for each $\tau = (h, r, t) \in \mathcal{G}$. Top-$K$ triples are then selected for LLM inference. A known limitation of this setup is that it ignores relation information, especially when multiple relations exist between a pair of entities. To mitigate this, we increase $K$ by 200 compared to the triple-level retriever and merge multiple parallel relations into a unified triple, e.g., $(h, r_1, t); (h, r_2, t) \to (h, r_1 \oplus r_2, t)$.

**ReG@Path.** We follow Eq. 31 for supervision. Building on Luo et al. (2023), we fine-tune LLaMA-2-7B as a planning module to generate plausible relation paths given a query $q$. These predicted relation paths are then mapped onto the KG using beam search to retrieve grounded reasoning paths. We use a beam width of $k = 20$ for GPT-4o and GPT-4o-mini, and $k = 10$ for LLaMA3.1-8B to reflect their respective capacities.

**Other details.** Retriever training for the `ReG@Triple` and `ReG@Entity` variants, which adopt lightweight MLP and GNN (PNA) architectures respectively, is conducted on an RTX 4090 GPU. Both retrievers are trained for a fixed number of 80 epochs, and we select the checkpoint with the highest retrieval recall on the validation set for downstream evaluation. The LLM-based retriever used in `ReG@Path` relies on LLaMA-2-7B and is trained on two NVIDIA RTX A6000 GPUs using LoRA (Hu et al., 2022) finetuning procedure on 8 epochs.

For the BFS-based chain expansion used in structure-aware reorganization, the maximum expansion length $L$ reflects the depth of reasoning allowed. We configure $L$ based on dataset characteristics: for the CWQ dataset, which requires complex multi-hop reasoning, we do not impose any upper limit on $L$; for all other datasets, we set $L = 2$ to balance path coverage and computational efficiency.

## G  PROMPT TEMPLATES

Here we provide detailed prompt templates used in both the LLM-guided supervision refinement stage (*c.f.*, Fig. 6) and the downstream QA stage (*c.f.*, Fig. 7). For both stages, we integrate a set of paths into the prompt. Specifically, for a $L$-length path $P := (\tau_1, \cdots, \tau_L)$ where $\tau_i := (h_i, r_i, t_i)$, we format it as a directional evidence chain of entities and their connecting relations, *i.e.*, $h_1 \to [r_1] \to t_1 \to [r_2] \to t_2 \cdots \to t_L$ following a natural left-to-right progression for fluent LLM reasoning.

---

**Input Prompts for LLM-guided Refinement**

**System:**
Given a question, an answer, and reasoning paths, select relevant paths for answering the question based on these criteria:
1. Relations in the selected path semantically align with certain question keywords;
2. Reject paths that contain only irrelevant details or demonstrate logical inconsistencies with question requirements.
Return all selected paths prefixed with 'ans:', one per line.

---

**User:** // ICL example
Paths:
Path 0.
boeing company → [spaceflight.rocket_manufacturer.rockets_manufactured] → saturn v rocket → [spaceflight.rocket.manufacturer] → North American Aviation
Path 1.
boeing company → [business.business_operation.industry] → Aerospace → [business.industry.companies] → North American Aviation
Path 2.
Little Joe → [spaceflight.rocket.manufacturer] → North American Aviation
Path 3.
North American Aviation → [aviation.aircraft_manufacturer.aircraft_models_made] → North American XB-70A
Path 4.
saturn v rocket → [spaceflight.rocket.height_meters] → 110.6
Path 5.
saturn v rocket → [spaceflight.rocket.country_of_origin] → United States of America
Path 6.
saturn v rocket → [spaceflight.rocket.mass] → 3038500.0
Path 7.
Aerospace → [business.industry.companies] → North American Aviation

Question:
What rocket manufacturer produced Little Joe and also collaborated with Boeing on making a rocket with mass over 2.916e+06?
Answer(s):
North American Aviation

---

**Assistant:** // ICL example
To identify the relevant reasoning paths, we need to focus on relationships that:
1) Confirm the manufacturer produced Little Joe, 2) Show collaboration with Boeing on a rocket, and 3) Provide information about the rocket mass.
Path 0 is valid because it establishes collaboration with Boeing (via Saturn V) and confirms North American Aviation as the manufacturer.
Path 2 is valid because it directly proves North American Aviation produced Little Joe.
Path 6 is valid because it provides information about the rocket mass (3,038,500 - 2.916e+06).
For irrelevant paths, Path 1/7 link Boeing and North American Aviation via Aerospace, which is indirect (no collaboration on making a rocket).
Path 3 discusses aircraft models, which is irrelevant to rockets.
Path 4/5 Describe background info of Saturn V (irrelevant to its mass).
Therefore, the relevant paths are:
ans: Path 0
ans: Path 2
ans: Path 6

---

**User:** // Sample to infer
Paths:
. . .
Question:
. . .
Answers(s):
. . .

Figure 6: Detailed prompt for LLM-based refinement used in our experiments.

---

**Input Prompts for Downstream QA**

**System:**
Answer the question using evidence chains from a knowledge graph. Return formatted answers, each on a new line and prefixed with 'ans:'.

---

**User:** // ICL example
Evidence Chains:
Chain 1.
Lou Seal → [sports.mascot.team] → San Francisco Giants → [sports.sports_team.championships] → 2012 World Series; 2014 World Series; 2010 World Series

Question:
What year did the team with mascot named Lou Seal win the World Series?

---

**Assistant:** // ICL example. Here the detailed explanation is optional.
To find the year the team with mascot named Lou Seal won the World Series, we need to find the team with mascot named Lou Seal and then find the year they won the World Series.
From Chain 1, we can see that Lou Seal is the mascot of the San Francisco Giants, and that San Francisco Giants won the 2010, 2012, and 2014 World Series.
So, the team with mascot named Lou Seal (San Francisco Giants) won the World Series in 2010, 2012, and 2014.
Therefore, the formatted answers are:
ans: 2010 World Series
ans: 2012 World Series
ans: 2014 World Series

---

**User:** // Sample to infer
Evidence Chains:
. . .
Question:
. . .

---

Figure 7: Detailed prompt for downstream QA used in our experiments.

Table 9: QA performance of ReG@Triple under DeepSeek-R1 inference.

| | WebQSP-Sub | | | | | CWQ-Sub | | | | |
|---|---|---|---|---|---|---|---|---|---|---|
| | Macro-F1 | Micro-F1 | Hit | Hit@1 | Avg.Tokens | Macro-F1 | Micro-F1 | Hit | Hit@1 | Avg.Tokens |
| w/o S(I) & S(II) | 78.18 | 56.23 | 96.41 | 89.22 | 660.36 | 66.78 | 53.42 | 84.38 | 73.07 | 1026.53 |
| w/o S(I) | 79.82 | 58.88 | 94.23 | 88.77 | 527.62 | 67.6 | 53.18 | 82.51 | 73.63 | 837.85 |
| ReG | 80.32 | 59.13 | 93.71 | 87.68 | 531.92 | 69.87 | 53.67 | 84.22 | 74.44 | 799.119 |

For downstream QA prompts, our experiments show that the explanation component in the ICL example is optional for stronger LLMs. Particularly, we omit the explanation part in ICL when using GPT-4o and reasoning-focused LLMs, *i.e.*, QwQ-32B and DeepSeek-R1, for downstream QA.

# H    ADDITIONAL EXPERIMENTAL RESULTS

# I    FUTURE WORK

Our pipeline is designed around the KGQA task and assumes the availability of entity-relation structured knowledge. Its generalization to other tasks or data modalities (e.g., vision-language) remains to be explored. The LLM-refined supervision signals may be not exactly the oracle one, and we primarily evaluate the effectiveness of such a LLM-refinement strategy using automatic metrics on QA performance, lack of human evaluations due to the high cost. For the formulation part, incorporating LLM's inherent sensitivity to structure and position bias mentioned in Sec. 3.3 to the reward definition in Sec. 3.1 remains a future work.

