# OpenReview forum: "Weak-to-Strong GraphRAG: Aligning Weak Retrievers with Large Language Models for Graph-based Retrieval Augmented Generation"
_ICLR.cc/2026/Conference — Submitted to ICLR 2026_

### Official Review · Reviewer_iAfW · 2025-10-27

**Soundness:** 2
**Presentation:** 3
**Contribution:** 2
**Rating:** 2
**Confidence:** 5

**Summary:**

This paper proposes Refined GraphRAG, which leverage refined retrieval graph to train a retriever for GraphRAG. Specifically, the authors first generate a multi-faced candidate with the shortest path, query neighbors and answer neigbors. Then, LLMs would refine the candidate graph to reduce the size. Finally, a retriever is trained based on the refined graph. Experimental results demonstrate the effectiveness of the proposed ReG.

**Strengths:**

1. It is reasonable that using better retrieval graph leads to better  performance.
2. The paper is well-written and easy to follow.

**Weaknesses:**

1. The proposed approach appears ad-hoc for multi-faceted candidate generation. The authors argue that previous methods relying on shortest paths suffer from a lack of reasoning signal. However, simply including the neighbors of the query and answer nodes does not adequately address this issues, particularly for multi-hop QA.

2. The LLM-Guided Candidate Refinement introduces unfairness in the comparison. This refinement step effectively performs part of the generation process, as the LLM may directly identify the correct answer for the query. As a result, comparing this approach with baselines that lack such refinement is not entirely fair.

3. The experiments do not convincingly demonstrate the generalizability of the proposed method to OOD KGs. Both CWQ and GrailQA are based on Freebase. I suggest that the authors include experiments on datasets with different underlying KGs, such as MetaQA, to strengthen their claims.

**Questions:**

Please refer to the weaknesses.

---

> ### Author Response · Authors · 2025-12-04
> **Response to Reviewer iAfW (Part 1)**
>
> Thank you for your suggestions and time in reviewing our paper. Please find our detailed responses below to your concerns.
>
> > W1. The proposed approach appears ad-hoc for multi-faceted candidate generation. The authors argue that previous methods relying on shortest paths suffer from a lack of reasoning signal. However, simply including the neighbors of the query and answer nodes does not adequately address this issues, particularly for multi-hop QA.
>
> **A1** We respectfully point out that the reviewer’s concern stems from a **misunderstanding** of the role of the “multi-faceted candidate generation.” We make point-to-point clarification as follows.
>
> First, we clarify that `Multi-Faceted Candidate Generation` mainly refers to the generation of candidate paths/subgraph **only for training the retriever**. During inference, the retriever does not have access to the answer entities and needs to generalize to extract the desired subgraph for LLMs to generate correct answers.
>
> **The candidate generation is not ad-hoc.** The construction of multi-faceted candidates is directly motivated by the analysis in Sec. 3.1 & 3.2: shortest-path heuristics suffer from incompleteness, because complex multi-hop KGQA often requires evidence **beyond the q-a shortest path** (aggregation, comparison, constraints, multi-entity bridging, etc.). Query-centric and answer-centric neighborhoods are therefore included to cover structural cues that shortest paths systematically miss. This is not ad-hoc, but a **principled consequence of the combinatorial incompleteness identified in our theoretical formulation**. Without this step, the candidate space would omit evidence required for multi-hop QA, making any refinement impossible.
>
> **LLM refinement provides the core reasoning signal.** On top of this completeness-oriented construction, the LLM performs the crucial refinement: **selecting a subgraph that more closely approximates the oracle reasoning subgraph.** The goal of multi-faceted generation is not to solve the multi-QA problem, but to **provide a sound and sufficiently rich basis** from which the LLM can extract a coherent, near-oracle subgraph.
>
> **Evidence that Multi-Faceted Candidate Generation assists multi-QA.** Figure 5(b) of the paper **directly confirms** this theoretical motivation: ReG achieves **significant and consistent improvements on multi-hop questions**, demonstrating that the combination of (i) coverage via multi-faceted construction and (ii) refinement via LLM reasoning is essential for multi-hop retrieval quality.

---

> ### Author Response · Authors · 2025-12-04
> **Response to Reviewer iAfW (Part 2)**
>
> > W2. The LLM-Guided Candidate Refinement introduces unfairness in the comparison. This refinement step effectively performs part of the generation process, as the LLM may directly identify the correct answer for the query. As a result, comparing this approach with baselines that lack such refinement is not entirely fair.
>
> **A2** In fact, our key contribution is to leverage LLMs’ feedback to refine the candidate generation. Nevertheless, we need to clarify that **using LLMs solely can not achieve good performance**,  even with frontier models. KGQA requires retrieving precise evidence from large KGs, which is fundamentally different from text-only QA and cannot be solved reliably by parametric knowledge alone[1].
>
> Below we provide the zero-shot performance of LLMs across three models (DeepSeek-R1, GPT-4o, GPT-4o-mini, Llama-3.1-8B) and two datasets (WebQSP, CWQ):
>
> | Model                    | WebQSP-F1 | WebQSP-Hit | CWQ-F1 | CWQ-Hit |
> |--------------------------|-----------|------------|--------|---------|
> | GPT-4o (zero-shot)       | 52.05     | 72.11      | 39.04  | 46.13   |
> | GPT-4o (ReG@Triple)      | 78.76     | 91.40      | 62.34  | 71.51   |
> | GPT-4o-mini (zero-shot)  | 46.95     | 67.69      | 32.48  | 40.10   |
> | GPT-4o-mini (ReG@Triple) | 77.98     | 92.87      | 60.55  | 67.66   |
> | Llama-3.1-8B (zero-shot) | 40.44     | 61.00      | 27.53  | 34.38   |
> | Llama-3.1-8B (ReG@Triple)| 69.91     | 87.39      | 51.24  | 65.02   |
> |DeepSeek-R1 (zero-shot)|	46.27	|68.98	|40.07	|49.05|
>
> As shown in the above table,
> - Zero-shot performance is **substantially inferior** compared with ReG@Triple, and actually, many baselines (as seen in Table 1 of the paper).
> - Our method improves all LLMs significantly, showing it is model-agnostic and addresses a genuine retrieval bottleneck.
> - Even the frontier reasoning LLMs, DeepSeek-R1, yields substantially inferior performance.
>
> Therefore, **retrieval is still indispensable** (as stated in the first two paragraphs of Sec 1), and improving retrieval quality remains an important and impactful research direction. From the ablation studies in Table 4 and 5, both methodology designs are critical to the final performance.
>
> > W3. The experiments do not convincingly demonstrate the generalizability of the proposed method to OOD KGs. Both CWQ and GrailQA are based on Freebase. I suggest that the authors include experiments on datasets with different underlying KGs, such as MetaQA, to strengthen their claims.
>
> **A3** We need to clarify that **the nature of the OOD split in GrailQA is to evaluate the OOD generalization performance, as shown in the title of their paper [2]**.
>
> **References**
>
> [1] Graph Retrieval Augmented Generation: A Survey, arXiv’2024.
>
> [2] Beyond iid: three levels of generalization for question answering on knowledge bases, WWW’21.

---

### Official Review · Reviewer_cVAD · 2025-10-30

**Soundness:** 3
**Presentation:** 3
**Contribution:** 3
**Rating:** 6
**Confidence:** 3

**Summary:**

This paper addresses the challenge of aligning weak retrievers with LLMs in graph-based RAG systems. The authors identify two key problems: (1) weak supervision signals from heuristic methods (e.g., shortest paths) that introduce spurious connections or miss critical evidence, and (2) misorganized representation of retrieved graph information. They propose ReG (Refined graph-based RAG), which uses LLM feedback to refine supervision signals and employs structure-aware reorganization to present retrieved information in logically coherent chains.

**Strengths:**

The paper clearly articulates the limitations of current graph-based RAG approaches. The method also achieves state-of-the-art performance across benchmarks.

**Weaknesses:**

1. Limited novelty in core techniques: While the combination is effective, the individual components are relatively standard. Using LLM feedback to filter/refine candidates is not new (acknowledged in related work). BFS-based chain expansion is a straightforward graph traversal technique. The main contribution appears to be the specific application to graph-based RAG rather than methodological innovation
2. No analysis of retrieval quality independent of QA performance (e.g., precision/recall of retrieved triples vs. oracle)

**Questions:**

Same as above

---

> ### Author Response · Authors · 2025-12-04
> **Response to Reviewer cVAD (Part 1)**
>
> Thank you for your time and valuable comments about our paper. Please find our responses below to your concerns.
>
> > W1 Limited novelty in core techniques: While the combination is effective, the individual components are relatively standard. Using LLM feedback to filter/refine candidates is not new (acknowledged in related work). BFS-based chain expansion is a straightforward graph traversal technique. The main contribution appears to be the specific application to graph-based RAG rather than methodological innovation
>
> **A1** We need to clarify that while the surface form of the two modules may resemble familiar operations, their role and design in **graph-based RAG** are substantively different from prior LLM-refinement works, which operate **almost exclusively at the document level**:
>
> **(1) Prior work does not address the core difficulty unique to graph-based RAG.** In graph-based RAG, as introduced in Sec 3.1, **it introduces non-trivial challenges due to the combinatorial candidate space, the irregularity of graphs, and the absence of ground-truth subgraphs**, unlike document RAG where candidates are regular, self-contained text chunks. Thus, directly applying LLM filtering or prompting strategies from document-level RAG is **ineffective and theoretically ill-posed**.
> Our Definition 3.1 is the first formalization of this subgraph selection problem as a black-box combinatorial optimization, and it motivates the need for principled refinement tailored to graph structure.
>
> (2) Our refinement mechanism is the first to **align graph retrievers with LLM reasoning preferences**. Existing LLM filtering methods do not consider graph irregularity or the representational mismatch between unordered triple sets (retriever raw output), and sequential, chain-based reasoning (LLM preference). Our method is the first to explicitly address this retriever-LLM misalignment, which is core to graph-based RAG.
> Theoretical framing and practical refinement together form a complete, graph-specific weak-to-strong pipeline, not a reuse of document-level techniques.
>
> (3) The chain re-organization module is **not a trivial BFS traversal**. While BFS is a standard graph operator, its role in our system is non-trivial: it **recovers coherent reasoning chains from scattered triples and provides a structure that LLMs can reliably utilize** (Sec. 4.2), mitigates overthinking and evidence fragmentation (as supported by Table 5), and enables consistent performance gains across models and datasets (Table 4).

---

> ### Author Response · Authors · 2025-12-04
> **Response to Reviewer cVAD (Part 2)**
>
> > W2 No analysis of retrieval quality independent of QA performance (e.g., precision/recall of retrieved triples vs. oracle)
>
> **A1** We need to clarify that the key challenge in graph-based RAG is the lack of a ground-truth subgraph. **Different from document RAG, the combinatorial candidate space naturally requires significantly more computational resources or human efforts to obtain the desired ground-truth subgraphs**.
> Therefore, **precision/recall against oracle** triples cannot be computed. This limitation is shared across prior graph-based RAG work (e.g., SubgraphRAG, G-Retriever), as none of these provide **ground-truth reasoning subgraphs**.
>
> **Accessible Retrieval-only metrics.** While the full oracle subgraph is inaccessible, the gold answers are known. We thus provide a meaningful retrieval-only metric: **answer recall within the top-K retrieved triples**, which reflects retrieval quality independent of QA performance. We report this metric for SubgraphRAG and ReG (ours) under varying dataset size for training the retriever (1%, 5%, …, 80%) and retrieval volumes (top-10, top-20, to top-200 triples), in the CWQ dataset.
>
> **Results.** The table below shows that, across all training budgets and retrieval volumes, **ReG consistently achieves higher answer recall**. The gains are particularly pronounced **when the retrieval volume is low** (e.g., 10-50). This demonstrates that the refined supervision produced by ReG substantially improves retrieval quality itself, which is independent of QA decoding.
> | training-data | method       | 10      | 20      | 50      | 100     | 200     |
> |:--------------:|:-------------|:------|:------|:------|:------|:------|
> | **1%**        | subgraphrag  | 25.2  | 32.68 | 44.42 | 54.03 | 61.65 |
> |               | ours         | 28.49 | 38.79 | 50.58 | 59    | 66.27 |
> |               | improvement  | 3.29  | 6.11  | 6.16  | 4.97  | 4.62  |
> | **5%**        | subgraphrag  | 42.07 | 49.67 | 60.78 | 67.73 | 72.74 |
> |               | ours         | 50.34 | 59.2  | 68.24 | 72.78 | 76.14 |
> |               | improvement  | 8.27  | 9.53  | 7.46  | 5.05  | 3.4   |
> | **20%**       | subgraphrag  | 52.06 | 59.11 | 67.57 | 72.44 | 76.83 |
> |               | ours         | 56.87 | 63.8  | 70.5  | 74.71 | 77.54 |
> |               | improvement  | 4.81  | 4.69  | 2.93  | 2.27  | 0.71  |
> | **40%**       | subgraphrag  | 54.57 | 61.48 | 69.27 | 73.8  | 76.7  |
> |               | ours         | 59.31 | 65.35 | 72.71 | 76.08 | 77.98 |
> |               | improvement  | 4.74  | 3.87  | 3.44  | 2.28  | 1.28  |
> | **60%**       | subgraphrag  | 55.34 | 61.69 | 69.51 | 74    | 77.1  |
> |               | ours         | 60.98 | 66.75 | 72.61 | 75.63 | 78.05 |
> |               | improvement  | 5.64  | 5.06  | 3.1   | 1.63  | 0.95  |
> | **80%**       | subgraphrag  | 56.64 | 62.98 | 70.04 | 74.13 | 76.81 |
> |               | ours         | 61.86 | 67.6  | 73.16 | 76.01 | 78.65 |
> |               | improvement  | 5.22  | 4.62  | 3.12  | 1.88  | 1.84  |

---

### Official Review · Reviewer_zMjD · 2025-10-31

**Soundness:** 2
**Presentation:** 4
**Contribution:** 1
**Rating:** 2
**Confidence:** 5

**Summary:**

This paper proposes ReG to improve traditional KGQA by using LLMs to refine weak supervision signals and train the retriever and to reorganize retrieved knowledge into logical chains. The core claim is that this aligns "weak" retrievers with "strong" LLMs. While the experimental results on KGQA benchmarks are solid, the paper suffers from fundamental conceptual and methodological problems that make it low-quality.

**Strengths:**

- It solves a practical problem since heuristic-based supervision like shortest paths for graph retrievers is noisy and misaligned with LLM reasoning.
- The results are promising.​​ The method demonstrates strong performance gains on traditional KGQA datasets (WebQSP, CWQ) and shows impressive data efficiency, matching SOTA performance with only 5% of training data.

**Weaknesses:**

- The title and claims are misleading. This is indeed an LLM-based KGQA paper, not GraphRAG. The community refers GraphRAG as a complete pipeline, similar to but more than RAG, that involves ​​constructing a graph from raw documents​​ and then retrieving from it. This work operates purely on ​​existing KGs​​ in a traditional LLM-based KGQA setting, let lone not comparing against real GraphRAG baselines, the paper addresses a much narrower problem than it claims.
- It is a very important prerequisite that we use KGs to enhance LLMs for unseen or difficult domain-specific scenarios. This arouses two major problems of this paper:
    - There lacks the zero-shot performance of LLMs. If LLMs could already achieve good accuracy, what is the advantage of this paper? This is also a concern to the entire KGQA task, it is not generalizable and applicable for LLMs nowadays. Therefore, the contribution of this paper is not enough.
    - Weak-to-strong is a good hypothesis but should not be static. LLMs are treated as oracle, but the feedback should be used to iteratively refine the signal for better loop. The design lacks enough consideration to make it a real 'weak-to-strong'.

**Questions:**

- What is the advantage of training a specialized retriever compared to directly using the powerful zero-shot LLM? also any comparisons?
- If the alignment is not iteratively achieved, how do you ensure LLM is a reliable oracle since we often aim to solve the hallucination and domain knowledge lacking problem in GraphRAG?

---

> ### Author Response · Authors · 2025-12-04
> **Response Reviewer zMjD (Part 1)**
>
> We thank you for your time and effort in reviewing our work. Please find our response to your question below.
>
> > W1 Scope of title and claims.
>
> **A1** We respectfully point out that the **notion of “GraphRAG” used in the review does not fully reflect the definition** adopted in recent graph-based RAG literature: the research community has converged on a broader and more principled definition that encompasses KG-based retrieval settings[1]. Our work lies within the **graph-based retrieval-augmented generation (graph-based RAG)** line of research, which is **broader than the document-to-graph GraphRAG pipelines** emphasized in some recent industry systems. In the academic literature, graph-based RAG commonly refers to LLM reasoning grounded on retrieved graph evidence, where the underlying graph may be (i) constructed from documents or (ii) provided as a structured KG. Our work follows the second setting.
>
> Indeed, a substantial body of graph-based RAG work operates **directly on existing knowledge graphs**, without requiring graph construction from raw documents. Examples include G-Retriever [2], GNN-RAG [3], RoG [4], GraphRAG-FI [5], ToG [6], SubgraphRAG [7],  all of which consider KG-based retrieval and LLM grounding as part of the graph-RAG paradigm.
>
> In addition, we also share the same objective as in [1], while we highlight the challenges due to the absence of the ground-truth subgraph $G^*$.
>
> Our contribution is therefore not to solve the document-to-graph construction as emphasized in some recent industry systems, but to tackle a **fundamental and well-recognized challenge within graph-based RAG: aligning weak graph retrievers with LLM reasoning preferences**, especially in the **absence of ground-truth retrieval supervision**.
>
> To avoid potential misunderstandings in the future, we will change our title a bit to ``Aligning Weak Retrievers with Large Language Models for Knowledge-Graph-based Retrieval Augmented Generation’’.
>
> > W2.1 There lacks the zero-shot performance of LLMs. If LLMs could already achieve good accuracy, what is the advantage of this paper? This is also a concern to the entire KGQA task, it is not generalizable and applicable for LLMs nowadays. Therefore, the contribution of this paper is not enough.
>
> **A2** **We respectfully disagree with the reviewer’s assumption** that `...the entire KGQA task, it is not generalizable and applicable for LLMs nowadays. Therefore, the contribution of this paper is not enough`, which **contradicts empirical evidence shown below**.
> We need to clarify that **using LLMs solely can not achieve good performance**,  even with frontier models. KGQA requires retrieving precise evidence from large KGs, which is fundamentally different from text-only QA and cannot be solved reliably by parametric knowledge alone[1].
>
> Below we provide the zero-shot performance of LLMs across three models (DeepSeek-R1, GPT-4o, GPT-4o-mini, Llama-3.1-8B) and two datasets (WebQSP, CWQ):
>
> | Model                    | WebQSP-F1 | WebQSP-Hit | CWQ-F1 | CWQ-Hit |
> |--------------------------|-----------|------------|--------|---------|
> | GPT-4o (zero-shot)       | 52.05     | 72.11      | 39.04  | 46.13   |
> | GPT-4o (ReG@Triple)      | 78.76     | 91.40      | 62.34  | 71.51   |
> | GPT-4o-mini (zero-shot)  | 46.95     | 67.69      | 32.48  | 40.10   |
> | GPT-4o-mini (ReG@Triple) | 77.98     | 92.87      | 60.55  | 67.66   |
> | Llama-3.1-8B (zero-shot) | 40.44     | 61.00      | 27.53  | 34.38   |
> | Llama-3.1-8B (ReG@Triple)| 69.91     | 87.39      | 51.24  | 65.02   |
> |DeepSeek-R1 (zero-shot)|	46.27	|68.98	|40.07	|49.05|
>
> As shown in the above table,
> - Zero-shot performance is **substantially inferior** compared with ReG@Triple, and actually, many baselines (as seen in Table 1 of the paper).
> - Our method improves all LLMs significantly, showing it is model-agnostic and addresses a genuine retrieval bottleneck.
> - Even the frontier reasoning LLMs, DeepSeek-R1, yields substantially inferior performance.
>
> Therefore, **retrieval is still indispensable** (as stated in the first two paragraphs of Sec 1), and improving retrieval quality remains an important and impactful research direction.

---

> ### Author Response · Authors · 2025-12-04
> **Response Reviewer zMjD (Part 2)**
>
> > W2.2 Weak-to-strong is a good hypothesis but should not be static. LLMs are treated as oracle, but the feedback should be used to iteratively refine the signal for better loop. The design lacks enough consideration to make it a real 'weak-to-strong'.
>
> **A3** We respectfully point out that the reviewer’s comment relies on **two items of misunderstanding that do not hold in our setting**: (i) that LLMs are treated as oracles, and (ii) that weak-to-strong requires iterative LLM loops. **Neither is the case**. To clarify this conceptual mismatch, we make the following point-to-point clarification:
>
> **1. LLMs are not treated as oracles.** We first clarify that **LLMs are not treated as oracles** in our method. Zero-shot results (W2.1) already show that LLMs cannot directly solve KGQA due to the fundamental limitation of parametric knowledge: it cannot cover large, domain-specific KGs. However, when candidate subgraphs are explicitly provided, LLMs can offer substantially more informative signals than heuristic rules such as shortest paths that more closely approximate oracle subgraphs. This is achieved due to the **reasoning ability** of LLMs, not memorization.
>
> **2. Why iterative LLM feedback is neither necessary nor feasible.** As formalized in Sec. 3.1 and Sec. 3.2, retrieving the optimal subgraph is a black-box combinatorial optimization problem over an exponentially large search space. Running iterative LLM-in-the-loop refinement would require the LLM to evaluate a rapidly growing number of subgraphs, this is computationally infeasible. Instead, our approach uses a **single-shot refinement**:
> - compress the candidate pool by **~95%** (Table 1),
> - let the LLM refine among **already small** candidates,
> - then train the retriever **iteratively** with gradient updates.
>
>
> **3. The source of “weak-to-strong”.** The bullet 2 above gives the first source of “strong”: the retriever becomes stronger through iterative learning, not through repeated LLM feedback. A second, equally important source of “strong” arises from structural alignment.
>
> As discussed in Sec. 3.3, raw retrieved subgraphs are poorly aligned with how LLMs perform reasoning. The re-organization module in Sec. 4.2 reconstructs coherent reasoning chains from unordered triples, removes structural noise, and produces evidence representations that LLMs can reliably use. This alignment step is what enables LLMs to turn refined supervision from the first source into strong downstream gains.
>
> Together, these components form a practical and theoretically motivated weak-to-strong pipeline without requiring iterative LLM-in-the-loop updates.
>
>
>
> **References**
>
> [1] Graph Retrieval Augmented Generation: A Survey, arXiv’2024.
>
> [2] G-Retriever: Retrieval-Augmented Generation for Textual Graph Understanding and Question Answering, NeurIPS’24.
>
> [3] GNN-RAG: Graph Neural Retrieval for Large Language Model Reasoning, ACL’25.
>
> [4] Reasoning on Graphs: Faithful and Interpretable Large Language Model Reasoning, ICLR’24.
>
> [5] Empowering graphrag with knowledge filtering and integration, EMNLP’25.
>
> [6] Think-on-graph: Deep and responsible reasoning of large language model on knowledge graph, ICLR’24.
>
> [7] Simple Is Effective: The Roles of Graphs and Large Language Models in Knowledge-Graph-Based Retrieval-Augmented Generation, ICLR’25.

---

### Official Review · Reviewer_bXzL · 2025-11-06

**Soundness:** 3
**Presentation:** 3
**Contribution:** 3
**Rating:** 8
**Confidence:** 5

**Summary:**

This paper uses LLMs to refine input graphs and then trains retrievers via supervised learning. Experiments show that ReG achieves state of the art performance using only 5% of the training data and transfers well to out of distribution KGs.

**Strengths:**

1.	Tackles an important and timely problem.
2.	Thorough experimental evaluation, including an analysis of the proposed “overthinking” problem.
3.	Maintains high performance even when trained on just 5% of the data.

**Weaknesses:**

1.	Some parts of the text are hard to follow and would benefit from rewriting for clarity.
2.	Key methodological details are omitted and should be added.
3.     The use of LLM to evaluate each candidate P is expensive.

**Questions:**

1.	Figure 1 does not make the ReG workflow clear. Please redraw the diagram to more explicitly show the end to end pipeline and the role of each component.
2.	The motivation for introducing Definition 3.1 is unclear, since prior work has already formalized graph based RAG. Please clarify how Definition 3.1 differs from or complements existing formalisms (e.g., Peng et al., 2024, "Graph Retrieval Augmented Generation: A Survey").
3.	For graph refinement, the query centric neighborhood construction is well described, but the paper does not explain how answer centric neighborhoods are built in practice—only that they enable comparisons across answer candidates using numeric or categorical attributes. How are answer centric neighborhoods generated when entities lack attributes in the KG? Please provide concrete procedures or fallback strategies for such cases.

---

> ### Author Response · Authors · 2025-12-04
> **Response Reviewer bXzL (Part 1)**
>
> Thank you for your time and positive recommendation of our work!
>
> > W1 & Q1 Clarity of the paper, such as Figure 1 of the  ReG workflow.
>
> **A1** Thank you for the suggestion! We have improved the clarity of the presentation of our workflow:
> The revised figure can be found in https://postimg.cc/5Xqj1Zn7 , where we simplified the notations and highlighted the subsequent steps in the workflow: When given a question $q$, we will first extract a candidate subgraph using heuristics, and then leverage a LLM to refine the candidate subgraph (Step 1.1) for training the retriever (Step 1.2). Once the retriever is trained, it is expected to identify the entities and relations of the desired subgraph. Then, we will reorganize the retrieved subgraph to better align with the LLMs (Step 2), in order to facilitate the generation of the correct answer (Step 3).
>
> > W2 & Q2 Definition and motivation for Definition 3.1.
>
> **A2** Thank you for the nice suggestion. The motivation for establishing Definition 3.1 is to **highlight the challenge in graph-based RAG, due to the lack of ground-truth retrieving paths**. Indeed, both Definition 3.1 and the definition in 4.1 of the graph-based RAG survey [1] aim to identify an optimal subgraph from the given graph. More specifically, we share the same objective to maximize the question-answering performance $p(a|q,\mathcal{G})$. [1] decomposes the objective as:
>
> $$p(a|q,\mathcal{G}) \approx p\_\phi(a|q,G\^{\*})p\_\theta(G\^{\*}|q,\mathcal{G})$$
>
> and consider the generator $p\_\phi$ and the retriever $p\_\theta$ are both tunable. In our case, we keep the generator fixed, i.e., a frozen LLM $p\_\phi$, and thus focus on the retrieval problem itself. The optimal subgraph is then naturally characterized by
> $$
> G\^\* = \text{argmax}\_{G\subseteq \mathcal{G}} p\_\phi(a|q,G).
> $$
> **This means that the LLM provides a scoring function over candidate subgraphs. The LLM therefore plays the role of a black-box evaluator of retrieval quality**, which enables us to formalize subgraph selection as a **black-box combinatorial optimization** problem. Definition 3.1 is thus meant to complement existing formulations by **making the retrieval difficulty explicit**. In particular, it highlights why finding $G\^\*$ is intrinsically challenging: the evaluator is available only through costly LLM calls, and the search space $2^{|\mathcal{G}|}$ is combinatorial.
>
> We have supplemented the following discussion to make Definition 3.1 more accessible to the readers.
>
> > W3 The use of LLM to evaluate each candidate P is expensive.
>
> **A3** We need to clarify that the LLM evaluation cost in our method is relatively cheap for threefold reasons:
>
> **(1) Evaluation is performed only once per training example, on a highly compressed candidate set**. As shown in Table 1, structural merging reduces the raw candidate pool by **about 95% on GrailQA**. Thus, the LLM only scores a small set of candidate chains per example, rather than the large graph.
>
> **(2) This step is offline and amortized**. LLM-based refinement is used only during training data preparation and does not appear at inference time. Therefore, the evaluation cost does not scale with model deployment or test-time usage.
>
> **(3) The refined candidates substantially reduce downstream computational cost.**
> In addition, we also provide some quantitative results as references for the reduced cost by our method:
> - **Training**:  Figure 4a shows that ReG with only **5%** training data achieves superior performance than the baseline counterpart with **80%** training data.
> - **Inference**: Table 5 exhibits that ReG reduces the reasoning token budget of frontier reasoning LLMs (e.g., DeepSeek-R1) by up to 30% across datasets, mitigating overthinking and reducing inference cost.
> Therefore, the LLM evaluation overhead is largely compressed, one-time, and fully amortized, while the refined supervision significantly reduces both training and inference cost, leading to a net reduction in overall computational budget.

---

> ### Author Response · Authors · 2025-12-04
> **Response Reviewer bXzL (Part 2)**
>
> > Q3 How are answer-centric neighborhoods generated when entities lack attributes in the KG? Please provide concrete procedures or fallback strategies for such cases.
>
> **A4** Thank you for pointing out this potentially confusing point. We need to clarify that answer-centric neighborhoods are constructed as the k-hop KG neighborhood (k typically set to 1) around an answer entity. This neighborhood includes relational edges, types, and **numeric or categorical attributes when such attributes exist**. Thus, these attributes are optional signals within the neighborhood, not prerequisites for generating $\mathcal{P}_a$.
>
> The original phrase in the paper (`enable comparisons via numeric or categorical attributes` in page 5) was intended to illustrate **one specific subclass of queries**, i.e., comparison/aggregation types (see details in Appendix E.1, page 19), where constructing answer-centric neighborhoods is necessary. It illustrates that $\mathcal{P}_a$  can capture attribute-based evidence when present, but was not meant to imply that our construction depends on these attributes. We will revise the wording to avoid this unintended implication.
>
> **References**
>
> [1] Graph Retrieval Augmented Generation: A Survey, arXiv’2024.

---

### Author Response · Authors · 2025-12-04
**Summary of Rebuttal (Part 1)**

Dear Reviewers and Area Chairs,


We are grateful to the reviewers for their time and constructive evaluations. We also extend our sincere thanks to the area chairs for managing the additional workload due to the OpenReview bug. Although the reviewers didn’t get the chance to reply to us before the bug period, we provide below a summary of our key contributions and replies to assist with the decision process.


## Contributions of this work

This work develops the first formulation of and highlights the challenges of the optimal subgraph retrieval in graph-based RAG. **Reviewers bXzL, zMjD, cVAD all agree with the importance of the problem**.
- Theoretically, we show that it is computationally intractable for the retriever to identify the optimal subgraph. The weak supervision guided by heuristics will introduce spurious signals that further mislead extraction. Hence, we propose to use LLMs to refine the weak supervision that significantly improves the quality and reduces the theoretical computational requirement for extracting the desired subgraph.
- Empirically, the representation of the subgraph fed to LLMs is also unorganized such that LLMs is struggling to fully leverage the retrieved information. Hence, we introduce a
- We conduct extensive experiments to demonstrate significant improvements of our method ReG. In addition, the improved supervision quality enables ReG to match the state-of-the-art performance with 5% training data and to transfer to out-of-distribution
KGs. Notably, when adopted to reasoning-based LLMs, ReG reduces the reasoning
token cost by up to 30% and improves the performance by up to 4%. **Especially, all Reviewers kindly recognized the empirical improvements**.

Initially, this work received two positive recommendations (bXzL:8, cVAD:6) and two negative recommendations (zMjD:2, iAfW:2). Nevertheless, we believe **there exist several factual misinterpretations by Reviewers zMjD and iAfW that can be directly addressed via our clarifications.**


## Questions and our responses

While we provided detailed point-to-point responses to each reviewer, we also summarized our responses to assist with the decision-making process.

### Relations with graph-based RAG, and refined RAG with LLM feedback (bXzL, zMjD, cVAD)

This work **shares the same objective as graph-based RAG [1], and focuses on improving the retrieval of the desired subgraph in graph-based RAG** to assist LLMs in generating correct answers. Detailed theoretical derivations can be found in our responses to Reviewers bXzL.

In fact, a substantial body of graph-based RAG work operates **directly on existing knowledge graphs**, without requiring graph construction from raw documents. Examples include G-Retriever [2], GNN-RAG [3], RoG [4], GraphRAG-FI [5], ToG [6], SubgraphRAG [7],  all of which consider KG-based retrieval and LLM grounding as part of the graph-RAG paradigm.

In our response to Reviewer cVAD, we also clarified that **retrieving the desired subgraph with LLM feedback introduces non-trivial challenges due to its combinatorial nature**, hence our work is not a direct extension of using LLM feedback to graph-based RAG.

---

> ### Author Response · Authors · 2025-12-04
> **Summary of Rebuttal (Part 2)**
>
> ### Can directly using LLMs address the problem? (zMjD, iAfW)
>
> **Directly using LLMs can not address the problem, even using frontier reasoning models**. Essentially, KGQA requires retrieving precise evidence from large KGs, which is fundamentally different from text-only QA and cannot be solved reliably by parametric knowledge alone. Below we provide the zero-shot performance of LLMs across three models (DeepSeek-R1, GPT-4o, GPT-4o-mini, Llama-3.1-8B) and two datasets (WebQSP, CWQ):
>
> | Model                    | WebQSP-F1 | WebQSP-Hit | CWQ-F1 | CWQ-Hit |
> |--------------------------|-----------|------------|--------|---------|
> | GPT-4o (zero-shot)       | 52.05     | 72.11      | 39.04  | 46.13   |
> | GPT-4o (ReG@Triple)      | 78.76     | 91.40      | 62.34  | 71.51   |
> | GPT-4o-mini (zero-shot)  | 46.95     | 67.69      | 32.48  | 40.10   |
> | GPT-4o-mini (ReG@Triple) | 77.98     | 92.87      | 60.55  | 67.66   |
> | Llama-3.1-8B (zero-shot) | 40.44     | 61.00      | 27.53  | 34.38   |
> | Llama-3.1-8B (ReG@Triple)| 69.91     | 87.39      | 51.24  | 65.02   |
> |DeepSeek-R1 (zero-shot)|	46.27	|68.98	|40.07	|49.05|
>
> As shown in the above table,
> - Zero-shot performance is **substantially inferior** compared with ReG@Triple, and actually, many baselines (as seen in Table 1 of the paper).
> - Our method improves all LLMs significantly, showing it is model-agnostic and addresses a genuine retrieval bottleneck.
> - Even the frontier reasoning LLMs, DeepSeek-R1, yields substantially inferior performance.
>
> Therefore, **retrieval is still indispensable** (as stated in the first two paragraphs of Sec 1), and improving retrieval quality remains an important and impactful research direction.
>
> **Instead, the refined supervision signal improved the retrieved quality, hence also the final performance**. In our response to Reviewer cVAD, we provided quantitative results showing the **consistent and significant improvements up to 9% in the quality of retrieved entities**.
>
> ### Function of Multi-Faceted Candidate Generation (bXzL, iAfW)
>
> First, we clarify that `Multi-Faceted Candidate Generation` mainly refers to the generation of candidate paths/subgraph **only for training the retriever**. During inference, the retriever does not have access to the answer entities and needs to generalize to extract the desired subgraph for LLMs to generate correct answers.
>
> **The candidate generation is not ad-hoc.** The construction of multi-faceted candidates is directly motivated by the analysis in Sec. 3.1 & 3.2: shortest-path heuristics suffer from incompleteness, because complex multi-hop KGQA often requires evidence **beyond the q-a shortest path** (aggregation, comparison, constraints, multi-entity bridging, etc.). Query-centric and answer-centric neighborhoods are therefore included to cover structural cues that shortest paths systematically miss. This is not ad-hoc, but a principled consequence of the combinatorial incompleteness identified in our theoretical formulation. Without this step, the candidate space would omit evidence required for multi-hop QA, making any refinement impossible.
>
> **LLM refinement provides the core reasoning signal.** On top of this completeness-oriented construction, the LLM performs the crucial refinement: **selecting a subgraph that more closely approximates the oracle reasoning subgraph.** The goal of multi-faceted generation is not to solve the multi-QA problem, but to **provide a sound and sufficiently rich basis** from which the LLM can extract a coherent, near-oracle subgraph.
>
> **Evidence that Multi-Faceted Candidate Generation assists multi-QA.** Figure 5(b) of the paper **directly confirms** this theoretical motivation: ReG achieves **significant and consistent improvements on multi-hop questions**, demonstrating that the combination of (i) coverage via multi-faceted construction and (ii) refinement via LLM reasoning is essential for multi-hop retrieval quality.
>
>
> ### Remaining concerns
>
> We provided clarifications on the potentially confusing points in our responses to Reviewers bXzL, zMjD, iAfW.
>
> **References**
>
> [1] Graph Retrieval Augmented Generation: A Survey, arXiv’2024.
>
> [2] G-Retriever: Retrieval-Augmented Generation for Textual Graph Understanding and Question Answering, NeurIPS’24.
>
> [3] GNN-RAG: Graph Neural Retrieval for Large Language Model Reasoning, ACL’25.
>
> [4] Reasoning on Graphs: Faithful and Interpretable Large Language Model Reasoning, ICLR’24.
>
> [5] Empowering graphrag with knowledge filtering and integration, EMNLP’25.
>
> [6] Think-on-graph: Deep and responsible reasoning of large language model on knowledge graph, ICLR’24.
>
> [7] Simple Is Effective: The Roles of Graphs and Large Language Models in Knowledge-Graph-Based Retrieval-Augmented Generation, ICLR’25.

---

### Meta-Review · Area_Chair_59hP · 2025-12-19

**Summary:**

This paper uses LLMs to refine input graphs and then trains retrievers via supervised learning. Experiments show that ReG achieves state of the art performance using only 5% of the training data and transfers well to out of distribution KGs.

Strength:

* Tackles an important and timely problem.
* well-written and easy to follow
* Maintains high performance even when trained on just 5% of the data.

Weakness:
* Fairness issues
* OOD generalization across different KGs

**Reviewer Concerns:**

**Concerns I consider largely resolved:**

* **Missing zero-shot comparison (zMjD/iAfW):** The authors added a table comparing Zero-shot vs. ReG for multiple models on WebQSP/CWQ, showing that Zero-shot performance is significantly lower than ReG.
* **Lack of retrieval-only metrics (cVAD):** The authors added the "recall of answer entity coverage by top-K triples" and provided a comparison table of SubgraphRAG vs. ReG. ReG performs consistently higher across different training ratios and top-K settings.
* **Positioning controversy (zMjD):** The authors provided a broader definition of graph-based RAG and promised to change the title to "Knowledge-Graph-based RAG" to avoid potential misunderstandings.

**Concerns I consider still outstanding (and the primary reasons for my Rejection):**

* **Fairness issues (iAfW):** The authors argued that "relying solely on LLMs for KGQA is insufficient," supporting this with zero-shot results. However, the reviewer's core concern remains: whether the **LLM's participation in refinement fundamentally alters the comparison setting**, and whether the method should be aligned with baselines utilizing **equivalent LLM resources**. The rebuttal mainly offered argumentation and clarification on this point, lacking stricter, resource-aligned comparative experiments.
* **OOD generalization across different KGs (iAfW):** The authors clarified the definition of the GrailQA OOD split, but the reviewer specifically requested empirical evidence based on **"different underlying KGs"** (e.g., MetaQA). No new experimental evidence was provided to close this loop.
* **Methodological Novelty (cVAD):** While the authors addressed the concern that the method is "not simply BFS," the overall contribution still leans towards **system integration**. Given the current split in reviews, this explanation is insufficient to overturn the overall judgment of the two strong rejects.

**Reviewer Scores:**

bXzL (8): Likely to maintain the score (the issues raised were mostly regarding writing and cost, which are fixable/minor revisions).

cVAD (6): Likely to remain a 6 (the addition of retrieval-only metrics makes their stance more solid, but they were already borderline/ambivalent).

zMjD (2): There is a possibility of a slight increase to 4(due to the promised title change and added zero-shot comparison), but their judgment regarding the validity of the "weak-to-strong" setting and the sufficiency of the contribution will likely remain negative.

iAfW (2): I expect the score to remain a 2; this is because the core concerns regarding fairness and cross-KG OOD generalization have not been fully resolved by the new experiments.

---

### Decision · Program_Chairs · 2026-01-26

Reject